# The Genes–Stemness–Secretome Interplay in Malignant Pleural Mesothelioma: Molecular Dynamics and Clinical Hints

**DOI:** 10.3390/ijms24043496

**Published:** 2023-02-09

**Authors:** Giulia M. Stella, Caterina Marchiò, Elia Bari, Ilaria Ferrarotti, Francesco R. Bertuccio, Antonella Di Gennaro, David Michael Abbott, Paola Putignano, Ilaria Campo, Maria Luisa Torre, Angelo G. Corsico

**Affiliations:** 1Department of Internal Medicine and Medical Therapeutics, University of Pavia Medical School, 27100 Pavia, Italy; 2Unit of Respiratory Diseases, Cardiothoracic and Vascular Department, IRCCS Policlinico San Matteo, 27100 Pavia, Italy; 3Department of Medical Sciences, University of Turin, 10124 Turin, Italy; 4Pathology Unit, FPO-IRCCS Candiolo Cancer Institute, Strada Provinciale 142, 10060 Candiolo, Italy; 5Department of Pharmaceutical Sciences, University of Piemonte Orientale, Largo Donegani 2/3, 28100 Novara, Italy; 6Department of Surgical, Pediatric and Diagnostic Sciences, University of Pavia, 27100 Pavia, Italy

**Keywords:** malignant pleural mesothelioma, genetics, microenvironment, targeted therapies

## Abstract

MPM has a uniquely poor somatic mutational landscape, mainly driven by environmental selective pressure. This feature has dramatically limited the development of effective treatment. However, genomic events are known to be associated with MPM progression, and specific genetic signatures emerge from the exceptional crosstalk between neoplastic cells and matrix components, among which one main area of focus is hypoxia. Here we discuss the novel therapeutic strategies focused on the exploitation of MPM genetic asset and its interconnection with the surrounding hypoxic microenvironment as well as transcript products and microvesicles representing both an insight into the pathogenesis and promising actionable targets.

## 1. Introduction

Malignant pleural mesothelioma (MPM) is a rare and extremely aggressive neoplasm that arises from the pleural mesothelium and has a pathogenetic trigger highly linked to asbestos exposure. To date, MPM cannot be irradicated with effective therapies, and the average survival of affected patients is approximately 12–15 months from diagnosis [1,2,3]. The incidence in Europe is expected to peak around 2025 due to the long latency that can elapse between exposure to asbestos fibers and the onset of the disease [4]. MPM diagnosis is generally reached through conventional morphology and immunohistochemistry (IHC) [5,6,7,8]. The expressions of BAP1, EZH2, and MTAP proteins are used to classify chronic pleuritis from a malignant disease [9]. However, in some instances, a differential diagnosis between MPM and carcinoma is not possible, and even an exhaustive ad excludendum IHC workup may be inconclusive(Figure 1).

Asbestos is a direct non-mutagenic carcinogen; when its biopersistent nanometric fibers are inhaled, they reach the pleural space and interact directly with cellular receptors, causing a chronic inflammatory response with inappropriate activation leading to cellular proliferation [10].

Furthermore, it is known that chronic exposure to asbestos induces an inflammatory response in the mesothelial microenvironment with an immunosuppressive character that contributes to neoplastic progression. In fact, the lack of blockable tumor targets and the presence of a heterogeneous peritumoral stroma that favors a malignant evolution explains the failure of the most modern biological antiproliferative drugs, including small molecules, monoclonal antibodies and, more recently, immunotherapies, which have revolutionized the therapeutic management of many other human tumors. This very complex context is thus characterized by the absence of “driver” somatic genetic alterations on the one hand and by the presence of genetic alterations at the germinal level, which are known to have a predisposing role in the development of disease. Moreover, genetic signatures reflect the Darwinian selection of MPM cells operated by the specific tumor-surrounding milieu. According to the revised cataloging of cancer hallmarks, it is well known that the factors of the microenvironment affect cell behavior according to chemical (e.g., growth factors, cytokines, nutritional status, the chemical composition of the matrix) and mechanical (e.g., mechanical stress, matrix stiffness) dynamics. It is also known that the malignant potential of cells occurs when they become able to disseminate from their natural environment into the lymphatic and blood vessels and colonize distant organs [11,12]. MPM is characterized by synchronous multiple pleural localizations and presents a peculiar metastatic pattern. In advanced disease stages, it is characterized by invasion of the lymph nodes, lung, and chest wall, but the invasion of the peritoneum and pericardium can occur as well. Only in advanced cases, and relatively rarely, distant spreading arises [13,14,15]. A deeper understanding of tumor–stromal crosstalk is mandatory to efficiently impact MPM progression and improve a patient’s outcome.

## 2. Genetic Alterations and Disease Stage

Although some data are available regarding inter-individual susceptibility to asbestos carcinogenic potential and MPM onset, and several signatures and polymorphisms have been discovered to influence the risk of developing the disease [for a review, see [16,17,18,19,20], fewer data are available regarding the genetic asset associated to disease progression and patient’s prognosis. Genes known to be altered and associated with MPM onset are detailed in Table 1 and Table 2. Notably, MPM can occur spontaneously in the absence of documented exposure to asbestos or other risk factors. The most relevant data regards the detection of alterations in the BRCA1-associated protein gene (BAP1) [21]. The *BAP1* gene encodes a deubiquitylase found to be associated with multiprotein complexes that regulate key cellular pathways, including the cell cycle, cellular differentiation, cell death, gluconeogenesis, and the DNA damage response. BAP1 behaves as a tumor suppressor gene whose mutations predispose to MPM onset. Low doses of asbestos are sufficient to trigger MPM in the presence of genetic predisposition. Loss of BAP1 protein expression is documented in >50% of cases [22]. Somatic BAP1 changes are frequently reported, followed by mutations in *NF2* (encoding for merlin) and CDKN2A (encoding for p16INK4A and p14ARF) genes. Germline mutations in BRCA1 associated protein-1 (*BAP1*) gene have been reported in families with a high MPM incidence; *BAP1* somatic alterations can coexist in a condition of biallelic inactivation. When mesotheliomas were acquired, both mutations that shortened the BAP1 gene and uncontrolled expression of BAP1 were found. Germline BAP1 mutations cause mesothelioma and other malignancies, namely, uveal cancer, meningioma, and melanoma, overall defined as “BAP1- related cancer syndrome”. Germline mutations affecting the BAP1 gene are inherited and exist in an autosomal dominant phenotype, with the first mutated allele being inherited and the second inactivating mutation being acquired later in life. Notably, patients with MPM due to the BAP1-related cancer syndrome seem to have a better prognosis [22,23,24]. Great efforts have been directed toward understanding the genetics behind what makes one more susceptible to the cancerogenic potential of asbestos. Particular focus has been placed on the genes involved in inflammatory infiltration, oxidative stress, chromosome instability, and response to treatments [25]. In this perspective, it has been reported that specific variants in three genes associated with iron metabolism, namely, ferritin, transferrin, and hephaestin, are significantly associated with protection against the development of MPM [26,27]. Moreover, different pathway signatures have been detected in MPM samples from patients in response to tissue damage after lung-sparing surgery, chemotherapy, and radical hemi-thoracic radiotherapy treatment with curative intent [28]. Due to their role in the inflammasome, polymorphisms of *NLRP1* and *NLRP3* genes have been hypothesized to be involved in determining genetic susceptibility to MPM [29]. However, preliminary results have not been validated in a cohort of MPM cases and the controls with known asbestos exposure in Northern Italy, and thus, further validation is required [30].

Several changes have been introduced in the last edition of the tumor, node, and metastasis (TNM) staging system for MPM as a better evaluation of tumor thickness and a refinement of the N classification as important factors associated with survival [31,32]. However, several limitations persist, and the TNM (as well as the systems proposed by the International Mesothelioma Interest Group—IMIG) system remains almost inadequate. The need to improve the accuracy of staging criteria is strictly related to the rarity of the MPM on the one hand, to its morphologic, biologic, and clinical heterogeneity on the other, and to the low percentages of cases that undergo surgical resection with a clear examination of the tumor by a pathologist. Thus, it makes sense to identify disease markers to be integrated with clinical features and validated into heuristic algorithms to efficiently assess disease progression. Great efforts have been made to identify genetic signatures with the predictive values of tumor aggressiveness and patient outcome. Cytogenetic studies performed on MPM have demonstrated heterogeneous and highly variable chromosomal aberrations, with only a few features being shared between patients. Loss-of-heterozygosity mainly occur in 1p, 3p, 6q, 9p, 13q, 15q, and 22q chromosomal regions. Two of these regions are most frequently altered, namely, the *CDKN2A–ARF* gene at 9p21 and *NF2* at 22q12, which behave as tumor suppressors [33]. Oncogene activation events encompass missense somatic mutation in *KRAS* and *TP53,* which have been detected in human MPM samples and are known to induce not only epithelial transformation but also aggressive mesotheliomas in animal models [34]. Transformation of human epithelial mesothelium is known to be regulated by ERKs transducers, which have different roles in the regulation of cell injury and repair, and a critical role is played by ERK2 [35]. Some genomic events are known to be associated with MPM progression. For instance, Ivanov et al. [36] demonstrated differential copy number alteration in patients with long- and short-term recurrence and identified chromosomal losses in 1p, 9p, 9q, 4p, and 3p, and gains in 5p, 18q, 8q, and 17q, as commonly shared by the analyzed cases. Deletion in 9p21.3 was specifically associated with a worse outcome. Another issue regards the hepatocyte growth factor (HGF)/scatter factor and its receptor tyrosine kinase, MET, which are highly expressed in most human malignant mesotheliomas. MET gene is located on chromosome 7q31. In the cancer context, MET phosphorylation is responsible for the activation of genetic programs defined as Invasive Growth (IG), which drives metastatic spreading [37]. Mutations in the semaphorin and in the juxtamembrane domains of the receptor and an alternative exon 10 splicing have been reported [38] in MPM. In addition, MET overexpression is frequently reported in MPM, mainly in the epitheliod forms, but this data is not directly related to MET gene amplification, and it is not fully associated with patient outcomes [39,40]. Thus, the predictive role of Met overexpression as a predictive marker for targeted drugs is not fully verified [41]. The activation of the HGF/MET pair is consistent with that of EMT (epithelial-to-mesenchymal transition), which is known to be induced by several transducers, among which transforming growth factor β (TGF-β). EMT- or IG derives from oxidative damages induced by asbestos on mesothelial cells and, once activated, promotes the acquisition by a transformed cell of more aggressive phenotypes and invasive capacity markers [42]. In this context, several mesenchymal markers are expressed and often redundant: phosphorylation SMAD1/5 pathway is essential to activate EMT by TGF-β [43], which can also act through the mitogen-activated protein kinase (MAPK) [44]. It has been shown that HGF-MET pair-mediated cell proliferation of human MPM cell lines is associated with the occurrence of activating mutations in the phosphoinositide 3-kinase (PI3K)/MAPK 5/Fra-1 pathway transducers, which mediate resistance to anti-MET agents [45,46]. We and others also confirmed the activation of the MAP/ERK pathway, which represents a potentially actionable target to impact MPM progression and aggressiveness. Therefore, a strong rationale to multi-kinase combination therapy persists [47,48,49]. The post-transcriptional analysis reported that the miR-200 family (mainly miR-141 and miR-200c) could impact TGF-β-induced EMT through reciprocal repression with zinc-finger E-box-binding homeobox 1(ZEB1) [50].

The advent of massively parallel sequencing techniques has revolutionized the field of molecular oncology, giving the opportunity to discover the entire cancer genome and identify novel targeted regions and epigenetic and RNA markers [51,52,53]. The first comprehensive genomic analysis applied to MPM confirmed BAP1, CDKNA2A/B, and NF2 as the most frequently mutated genes [54,55]. On the other hand, transcriptomic profiling revealed different subgroups closely but strictly recalling known histologic subtypes, with the mesenchymal one, characterized by the activation of the EMT process, being associated with sarcomatoid morphology and, as expected, with more aggressive clinical behavior [56]. Subsequent and recent works pointed out changes affecting p53/DNA repair and PIK3CA pathways as being associated with reduced overall survival [57,58], demonstrating an overall poorly mutated landscape with high levels of tumor mutational burden (TMB) in only 5% of cases [59]. Different genetic signatures (including CDH2, CKS2, KIF11, KIF88, Lox, NF2, TP53, SETD2, LATS2, SETDB1, PBRM1, LATS1, SETD5) have been more recently proposed as independent prognostic tools in MPM [60,61]. Very recently, Zhang and coll. [62] were able to demonstrate that the BAP1 event occurs as early as clonal selection, whereas changes in NF2 genes, leading to Hippo inactivation, are selected later in cancer progression or evolve in parallel with the evolutionary trajectory. The same group also identified lesions affecting chromosome 4 as mutations in the FBXW7gene (4q31.3) as negative prognostic tumor suppressors involved in resistance to antimicrotubule agents, such as vinorelbine. Overall, MPM evolution shapes the MPM microenvironment since unstable genomic clusters generate more efficient immunosurveillance, ultimately leading to a scarce response to immune checkpoint inhibitors.

In conclusion, it is well known that MPM has a uniquely poor somatic mutational landscape and that the disease is mainly driven by selective pressure on the microenvironment. Thus, available therapies are not MPM-specific, and patient outcome is still poor and modestly affected by current treatments. The absence of driving somatic lesions explains the clinical failure of small molecules, whereas immunotherapy has shown limited advantages in association with standard chemotherapy. Growing interest is addressed to the germline MPM profile. A better understanding of the genetic variants and polymorphisms in MPM patients will help decipher a still unexplored milieu and improve mechanistic knowledge of MPM biology and interindividual susceptibility to asbestos. Moreover, this knowledge could be helpful in identifying novel actionable targets and in designing personalized and more efficient therapeutic strategies.

## 3. The Role of Microenvironmental Hypoxia

Hypoxia defines the condition in which tissue oxygen is available at an insufficient level to maintain and guarantee homeostasis. During malignant transformation, hypoxia develops as a consequence of a series of events, mainly rapid and uncontrollable cell proliferation, altered metabolism, and inappropriate vascularization of the tumor mass. Hypoxia regulates the crosstalk between tumor and microenvironment (TME) and is associated with worse patient prognosis. It represents a relevant target to impair tumor progression and chemoresistance as well [63,64,65,66]. The effects of asbestos on the extracellular matrix sustain MPM cells, preventing apoptosis and facilitating their spreading. Indeed, biopersistent fibers impact the tumor microenvironment, and MPM cells are known to produce collagen and matrix metalloproteinases, activate inflammatory cells and cytokines, and secrete angiogenetic factors, such as VEGF which acts as an autocrine growth factor and mitogen for malignant cells [67]. Angiogenetic agents have been tested in MPM as monotherapy [1], even though with unsatisfactory results. The hypoxic microenvironment is mainly responsible for the tumor’s neoangiogenic response but also modulates gene and microRNA (miRNA) expression, translational response pathways, and protein activation, ultimately contributing to chemo–radio-resistance through a variety of mechanisms [68]. Hypoxia is a hallmark driving force for tumor progression in solid cancers. The aggressive malignant potential of MPM cells, defined by increased clonogenic capacity and motility in the absence of proliferative gain, is increased in hypoxic conditions due to upregulation of the HIF1alfa, HIF2alfa, and target Glut-1 genes [69]. Moreover, hypoxia induces a metabolic switch in cancer cells leading to increased glucose uptake and the switch from pyruvate to lactate [70]. This phenomenon has been demonstrated in MPM as well through PET/CT scan analysis with 2-[19 F]-fluoro-2-deoxy-D-glucose (F-FDG) tracers. In vivo, F-FDG uptake in pleural MPM shows high correlations with upregulation of GLUT1, HIF1, VEGF, CD34, Ki67, and MTOR upregulation, and poor patient prognoses; HIF-1 activation increases glucose transport (via GLUT-1) as well as glutamine and L-type amino acid transport (via LAT1) in pleural MPM [71,72,73]. Moreover, hypoxia is also known to promote the Invasive Growth program as a consequence of the activation of the MET-promoter gene by HIF-1alfa [74]. The occurrence of von Hippler Lindau mutations detected by massively parallel sequencing of MPM samples has been reported to be associated with resistance to the HIF1alfa inhibitor YC-1 [75]. Moreover, important crosstalk between hypoxia-induced milieu and inflammatory pathways is sustained by the interaction between HIF1alfa and NF-ƙB transcription factors, a well-recognized regulator of several genes involved in inflammatory and immune responses [76,77]. Notably, a series of miRNAs whose expression is significantly deregulated by hypoxia, specifically, Let-7c-5p and miR-151a-5p, have been shown to be related to hypoxia and energy metabolism, differentially expressed in MPM and healthy mesothelium, and associated to worse clinical outcome [78]. Hypoxia is also related to chemoresistance in MPM according to several mechanisms: (i) reduction of expression of proton-coupled folate transporter (PCFT), which is associated with response to pemetrexed [79]; (ii) induction of expression on MPM cells of stemness (CD26, CD44, and ABCG22); iii) and hypoxia adaptation (ABCG2, ALDH1a1, HIFs) markers [80], even though the level of HIF1alfa seems to not predict patient survival [81]. In conclusion, as in other cancers, hypoxia deeply characterizes MPM’s surrounding stroma and is strongly associated with tumor onset and progression: it is also implicated in the failure of standard treatments. A deeper understanding of the molecular basis of the hypoxic mesothelioma microenvironment could be of help in the clinical setting as well.

## 4. The Hypoxia and Stemness Interconnection in MPM

Solid tumors contain a cellular subset that displays stemness properties: the cancer stem cells (CSC). Stem cells are defined as those elements which harbor the capacity to self-renew and the potency to generate differentiated cells [82]. The concept of CSC implies that tumor growth, similar to healthy tissue regeneration, is orchestrated by a small fraction of dedicated stem cells. The canonical theory describes stem cells and CSC as quiescent elements featuring intrinsic properties and capable of asymmetric division, thus giving rise to one stem cell and one rapidly defining cell. These cells also harbor the potential to sustain tumorigenesis and constitutively express molecular markers of multidrug resistance; they feature a quiescent, not-addicted phenotype, theoretically insensitive to chemo and targeted therapies [83,84,85]. According to this model, non-stem malignant cells have limited functional plasticity. More recent data support a more dynamic cellular hierarchy according to which CSC is not definitely rare or quiescent; they should express a vast range of gene products rather than a specific signature and are instructed by the signals of the niche [86,87]. The niche is a defined unique anatomic locus devoted to maintaining stem cell homeostasis. Moreover, it cooperates with stem cells in tissue regeneration and repair. The dialogue between stem cells and their niche is necessary for tissue homeostasis. The alteration of this balance or a deficient niche function is implicated in different pathologies [88,89,90,91,92,93,94]. In a similar fashion, CSCs reside in niches that also sustain CSC immune escape and are implicated in metastatic potential [95]. In this context, hypoxia is known to play a key role in maintaining CSC. Hypoxia is a key feature of MPM metabolism, related to oxygen consumption from rapidly proliferating cells, stromal reaction to biopersistent fibers, and neoangiogenesis [96,97]. In MPM patients, hypoxic areas are clearly detectable with F-fluoromisonidazole (FMISO) Positron Emission Tomography (PET) scans, and they are associated with increased metabolic activity on Fluorodeoxyglucose (FDG)-PET [54]. Hypoxia is also traced by immunohistochemistry with positive staining for a Hypoxia-Induced Factor 1α (HIF1α) [81]. The hypoxic microenvironment sustains undifferentiated phenotype and promotes invasive cellular behavior characterized by the projection of pseudopodia and increased expression of epithelial-to-mesenchymal transition markers such as E-cadherin, vimentin, and Bcl2 [69]. Coherently, with respect to histotype, sarcomatoid and biphasic MPM harbor enriched stem compartments, which are responsible for increased aggressiveness and chemoradioresistance if compared to epitheliod activation [98]. HIFs also regulate the stemness of CSCs, since they require activation of HIF-1α and HIF-2α to maintain their self-sustainability under hypoxic conditions [99,100,101]. Moreover, the hypoxic microenvironment may correlate with an altered immune response, strongly unbalanced toward immunosuppression which contributed to the intrinsic resistance of MPM to immune checkpoint inhibition in MPM [68,102]. Hypoxia, by acting via increased HIF1α-expression, induces PD-L1 expression in tumor cell lines as well as in murine macrophage and dendritic cells. In MPM, although being related to higher response rates to nivolumab [103,104,105], the increased PD1 and PD-L1 expression are mainly associated with sarcomatoid morphology and do not significantly affect overall survival [106,107,108]. This observation clearly points out that other parameters are involved with PD1 and PD-L1 in determining real sensitivity to immunotherapy in a complex and heterogeneous context as MPM, such as tumor mutational burden or tumor-infiltrating lymphocytes, the tumor-associated macrophages (TAMs), which are abundantly present in the MPM microenvironment and play an important role in inducing T-cell suppression, Tim-3 (T cell immunoglobulin and mucin-domain containing-3), which is a co-inhibitory receptor expressed on IFN-γ-producing T cells, FoxP3+ Treg cells, and innate immune cells (macrophages and dendritic cells), T-regulatory cells [88,109,110,111,112]. Moreover, the expression of PD-L1 in cancer stem cells has been related to immune evasion [113,114]. The knowledge of CSC in other solid cancers has led to the development of studies focused on the identification of initiating cancer cells in MPM as well [115]. Previous work reported the expression of MPM cells of several markers, such as SP, CD9, CD24, CD26, BMI1, OCT4, and NOTCH1, as related to a primary stem signature. Their expressions were correlated with several cancer-related genes, and phosphorylation of ERK by EGF was regulated by the expression of CD26 but not CD24 [116,117]. Moreover, the expression of Bmi-1+, uPAR+, and ABCG2+ has been associated with putative stemness and induction of resistance to platinum and pemetrexed in MPM cell lines [118]. The CSC compartment in MPM is emerging as a novel actionable target and mesospheres as a reliable model which recapitulates tumor onset and progression and can be exploited for therapeutic purposes [108,119]. The strategies developed essentially aim at: (i) destroying CSC; (ii) inducing differentiation of the stem elements; (iii) targeting the niche [120]. Several trials have been designed to impair MPM stem cells. The dual FAK and the Proline-rich tyrosine kinase 2 (Pyk2) member inhibitor, defactinib, was revealed as an efficient suppressor of CSCs [121,122,123]. This data supported the rationale for the phase II double-blind, randomized COMMAND-A study with defactinib in mesothelioma (Clinical trial NCT01870609). The results of the study didn’t show a beneficial effect of defactinib in improving OS (overall survival) and PFS (progression-free survival) in MPM [124]. However, the study pointed out two relevant issues: first was the potential role of the ezrin-radixin-moesin (ERM) protein (merlin or neurofibromin) which shares a regulatory domain with FAK-related Pyk2 kinases [125]. The NF2 gene encoding for the merlin protein is frequently mutated or inactivated in MPM [126,127,128] and represents a potential therapeutic target [129,130,131]. Notably, merlin is known to be critical for maintaining normal structure and function of the hematopoietic stem cell niche [132], and FAK cooperates in preserving the self-renewal of CSC, and the overexpression of merlin has been shown to affect proliferation and viability of CSC-enriched MPM [133]. Mithramycin is an antineoplastic agent produced by Streptomyces plicatus which behaves as DNA/RNA polymerase inhibitor, DNA-binding transcriptional inhibitor and antibiotic and has been observed to facilitate tumor necrosis factor (TNF)-α activity and Fas-ligand-induced, which is known to induce malignant cell reprogramming, differentiation, and senescence [134,135]. The NCT02859415 is a phase I/II trial aimed at evaluating continuous 24h intravenous infusion of mithramycin in patients with thoracic malignancies, including MPM. The study was recently terminated, and data are under analysis (https://clinicaltrials.gov/ct2/show/NCT02859415, accessed on 30 January 2023).

In conclusion, hypoxia represents a key feature involved in MPM onset and progression, and it is strictly related to the maintenance of cancer cell hierarchical compartments. However, the increasing body of knowledge regarding this peculiar bio-molecular context could be of help in designing tailored and more efficient therapeutic platforms.

## 5. Trancriptome and Secretome Profile of MPM

Strong preclinical evidence supports the role of hypoxia and MPM CSCs in determining disease resistance to therapies. A main mechanism is related to the alteration of microRNAs (miRNAs) [136]. The latter are small non-coding RNA (ncRNA) molecules that are known to play a regulatory role in cancer by acting either as oncogenes or tumor suppressors; moreover, they behave as actionable targets with predictive and prognostic roles. Other forms of ncRNAs include long non-coding RNAs (lncRNA), PIWI-interacting RNAs (piRNA), small interfering RNAs (siRNA), and microRNAs (miRNA) [137,138]. It is well known that the hypoxic microenvironment in MPM interferes with a range of miRNAs and that some miRNAs target HIF1α [139,140,141,142]. With respect to MPM, published data point out that let-7c-5p and miR-151-5p can be considered “hypoxamiRs” involved in tumor initiation and EMT maintenance, altered metabolism, chemoresistance, and poor prognosis [78]. On the other hand, several miRNAs are known to be downregulated in hypoxic conditions: miR-15b, 16, 19a, 20a, 20b, 29b, 30b, 30e-5p, 101, 141, 122a, 186, 320, and 197 [143]. Among them, miR-16 behaves as a tumor suppressor in MPM, being a promising actionable target to impair MPM growth [144,145]. TargomiRs are minicells (EnGeneIC Dream Vectors) loaded with miR-16-based mimic microRNA (miRNA) and directed against EGFR that are designed to counteract the loss of the miR-15 and miR-16 family miRNAs [146]. The MESOMIR 1 phase I trial showed a safe profile and demonstrated some therapeutic activity in a clinical setting in MPM (NCT02369198). In other cancer types, miRNAs have been reported to be involved in maintaining the stemness phenoptype of CSCs [147,148,149,150], but this role in MPM is not yet documented.

Extracellular vesicles (EVs) are small, membrane-bound structures that are released by cells into the extracellular space and are reported to be involved in mesothelial transformation [151,152,153]. Comprising exosomes, microvesicles, and apoptotic bodies, EVs play a crucial role in various physiological processes, including cell-to-cell communication and the transport of molecules between cells [154]. Recent studies have also uncovered that EVs have a significant impact on the development and progression of tumors [155]. Tumor-derived EVs are indeed found in high concentrations in the blood and urine of cancer patients [156] and contain a variety of biomolecules, including proteins, lipids, and nucleic acids, that can influence the behavior of surrounding cells [157]. Despite being limited in number, some publications suggest that EVs play a role in the initiation and progression of malignant pleural mesothelioma (MPM) [158]. Regarding MPM initiation, it is well established that asbestos exposure is a key cause, and EVs may play a role in the underlying biology. A study by Munson and colleagues found that asbestos-exposed lung epithelial cells and macrophages secrete EVs with unique proteomic cargo, including proteins, such as plasminogen activator inhibitor 1, vimentin, thrombospondin, and glypican-1 [159]. These EVs were also found to alter the gene expression of mesothelial cells, leading to changes related to epithelial-to-mesenchymal transition and other cancer-related genes, which contribute to disease outcomes. In another study, the same group exposed mice to asbestos by oropharyngeal aspiration, and 56 days later, plasma EVs were analyzed for proteomic content, revealing an abundance of acute-phase proteins, such as haptoglobin, ceruloplasmin, and fibulin-1 [160]. All of them were previously reported as implicative of asbestos exposure [161,162]. Regarding tumor progression, EVs play a significant role by promoting the growth and spread of cancer cells [163]. For MPM, EVs can carry oncogenes, which are genetic mutations that drive cancer development, and transfer them to other cells, leading to the formation of new tumors. For example, cargo can be transported between cells through actin-based cellular extensions, known as tunneling nanotubes, and MPM-EVs have been shown to increase their formation [164]. Furthermore, EVs can promote the growth and survival of cancer cells by stimulating angiogenesis and activating signaling pathways that promote cell proliferation. Studies investigating the protein and genetic (miRNA) content of EVs derived from MPM support these claims. Greening and colleagues used quantitative proteomics to investigate MPM-EVs and identified protein networks associated with angiogenesis and metastasis [165]. Furthermore, using iTRAQ^®^ mass spectrometry, it was found that MPM-EVs were enriched in proteins involved in carbon metabolism, the stress response to amino acids biosynthesis, protein processing in the endoplasmic reticulum, and antigen processing and presentation [166]. Other studies reported that MPM-EVs are rich in cytoskeleton proteins and their associated proteins (like moesin, ezrin, desmoplakin, actinin-4, and fascin [167]), signal transduction-involved proteins [15]. It’s particularly interesting that developmental endothelial locus-1, which can act as an angiogenic factor, has been found to be present and, thus, increase vascular development in the neighborhood of the tumor [168]. Additionally, MPM-EVs contain miRNAs involved in the post-transcriptional regulation of genes and protein expression, affecting cellular processes like proliferation, migration, and apoptosis. Specifically, higher levels of miR-16-5p [169], miR-222-3p, miR-30a-5p, and miR-16-5p were upregulated in exosomes from MPM, while miR-31-5p was significantly decreased [170]. EVs also play a crucial role in the immune response to tumors. Recent research has shown that MPM-EVs contain a high concentration of molecules that aid in antigen presentation, as well as immunoglobulins and complement factors [151,152]. Proteomic studies have also revealed the presence of protein networks that play a role in immunoregulation [149]. As a result, tumor-derived EVs can suppress the activity of immune cells, making it harder for the body to fight cancer. Additionally, EVs can help cancer cells evade the immune system by disguising themselves as healthy cells. In this regard, Clayton and colleagues have proven the role of tumor EVs bearing NKG2D in inhibiting immunological functions, thereby contributing to cancer immune evasion. Indeed, incubating MPM-EVs with leukocytes resulted in a marked reduction in the proportion of NKG2D-positive CD3+CD8+ cells and CD3− cells [171]. On the other hand, MPM cells also release exosomes that express a distinct set of proteins involved in antigen presentation, making them a valuable source of tumor-associated antigens. Mahaweni et al. have investigated the use of MPM-derived exosomes in dendritic cell-based immunotherapy, resulting in improved survival of tumor-bearing mice [172]. It is worth noting that this is a significant advancement, as MPM has few tumor-associated antigens. The potential to serve as a diagnostic tool for early cancer detection is another characteristic of EVs in MPM. Recent research has yielded promising results, though further validation with larger sample populations is needed. For instance, a study by Javadi and colleagues found higher ratios of biomarkers, such as galectin-1, mesothelin, VEGF, and osteopontin, in MPM samples compared to benign samples [173]. On the other hand, exosomal angiopoietin-1 levels were found to be higher in benign samples than in malignant ones. Additionally, changes in miRNA expression have been identified as potential diagnostic tools. For example, a study by Cavalleri and colleagues found that miR-103a-3p and miR-30e-3p provided the most discriminating combination when comparing plasmatic EVs from MPM and cancer-free patients [174]. Finally, the unique molecular composition of EVs presents an opportunity to utilize them as a means of delivering drugs and genetic material directly to tumor cells, ultimately leading to the development of targeted therapeutics. One of the major benefits of using EVs as a drug delivery system is their ability to selectively target specific cells and tissues, a process known as “homing”; another benefit is their ability to cross biological barriers [175]. Tumor-derived EVs, in particular, have been found to possess the ability to target cancer cells, thus delivering therapeutic agents directly to them while avoiding healthy cells [176]. This happens by recognizing specific receptors on the surface of cancer cells and has been demonstrated even in studies of MPM. In this regard, as reported by Monaco and colleagues, the transfer of microRNA-126 via exosomes has been shown to inhibit angiogenesis and cell growth in diseased tissue [156]. It is thus clear that EVs play a significant role in the development and progression of MPM [177]. Most importantly, EVs not only represent a key mechanistic system in MPM but also identify a powerful, actionable target. Further understanding of the mechanisms by which EVs contribute to cancer will pave the way for creating new diagnostic tools and therapeutic strategies.

## 6. Conclusions

The complex tumor and microenvironmental cross-talk that specifically characterizes MPM is becoming a novel promising therapeutic target. A main issue regards the role of hypoxia in inducing activation of proliferative signals and in maintaining neoplastic cell hierarchy and stem compartment. Hypoxia also interferes with MPM cell transcripts and miRNA expressions. The latter, together with exosomes and microvesicles, although behaving as key pathogenic clues, also define a possible line of therapy against MPM, featuring a powerful translational potential for future clinical development and use.

## Figures and Tables

**Figure 1 ijms-24-03496-f001:**
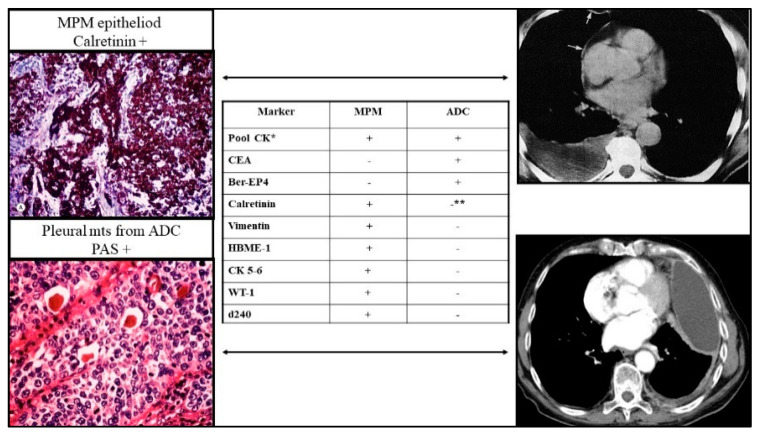
Immunohistochemistry workup in case of pleural masses and suspect of MPM. The study should include the expression of vimentin 8, TTF1, p40, calretinin, WT-1, D240, citokeratin 5/6, vimentin, S100, BER-EP4, CEA, CD31, CD34, desmin, and myogenin D1. The most relevant differential diagnosis regards MPM (e.g., calretinin positive) vs. pleural metastases from lung cancer (e.g., PAS-positive staining in case of adenocarcinoma -ADC); in some cases, results might not solve the origin of neoplastic cells proliferating into the pleural space. Moreover, differential diagnosis requires the exclusion of undifferentiated MPM, the presence of a small/dormant lung primary carcinoma, a pleural localization of melanoma (expression of S100), rhabdomyosarcoma (expression of desmin and myogenin D1), and angiosarcoma (expression of CD31, CD34), ectopic lung epithelial cells which undergo a malignant transformation, or of epithelial cancer from an unknown primary site of origin. **= possible, * = rare.

**Table 1 ijms-24-03496-t001:** Most commonly altered genes in MPM.

Gene	Location	Pathway	Variant Types
*BAP 1*	3p21.1	BRCA-associated protein-1 (ubiquitin carboxy-terminal hydrolase) (BAP1) is a gene that encodes a protein that has a high affinity for the BRCA1 protein and functions as a tumor suppressor protein	Missense mutations, nonsense mutations, silent mutations, frameshift insertions and deletions, and in-frame insertions and deletions.
*NF2*	22q12.2	Neurofibromin 2 (NF2) is a gene that encodes a protein that functions by connecting cytoskeletal components with cell-surface proteins, cytoskeletal proteins, and ion transport proteins.	Fusions, missense mutations, nonsense mutations, silent mutations, frameshift deletions and insertions, and in-frame deletions and insertions.
*CDKN2A*	9p21.3	Cell cycle control. Cyclin-dependent kinase inhibitor 2A (CDKN2A) gene encodes several protein isoforms that function as inhibitors of CDK4 and ARF.	Missense mutations, nonsense mutations, silent mutations, in-frame deletions, frameshift deletions and insertions, and whole gene deletions.
*CDKN2B*	9p21.3	Cell cycle control. Cyclin-dependent kinase inhibitor 2B (CDKN2B, also known as p15) is a gene that encodes a protein that binds to CDK4 or CDK6 and inhibits their activation.	Missense and silent mutations.
*TP53*	17p13.1	Cell cycle control.Tumor protein p53 (*TP53*) is a gene that codes for a tumor suppressor protein, cellular tumor antigen p53. The protein regulates expression of genes involved in cell cycle arrest, apoptosis, senescence, DNA repair, and changes in metabolism.	The most common alterations in TP53 are TP53 Mutation (32.56%), TP53 Missense (26.61%), TP53 c.217-c.1178 Missense (26.50%), TP53 Exon 5 Mutation (9.30%), and TP53 Exon 8 Mutation (8.49%) [3].
*SETD2*	3p21.31	Chromatin remodeling/DNA methylation. SET domain containing 2 (SETD2) is a gene that encodes a protein that is a member of a class of huntingtin interacting proteins. The protein functions as a histone methyltransferase specific for lysine-36 of histone H3.	Missense mutations, nonsense mutations, silent mutations, frameshift deletions and insertions, and in-frame deletions.
*PBRM1*	3p21.1	Chromatin remodeling/DNA methylation. Polybromo 1 (PBRM1) is a gene that encodes a protein that is a member of a protein complex that functions in ligand-dependent transcriptional activation by nuclear hormone receptors.	Missense mutations, nonsense mutations, silent mutations, frameshift deletions and insertions, and in-frame deletions.
*KMT2D*	12q13.12	Chromatin remodeling/DNA methylation. Lysine (K)-specific methyltransferase 2D (KMT2D) is a gene that encodes a protein that functions as a histone methyltransferase that methylates the LYS-4 position of histone H3.	Missense mutations, nonsense mutations, silent mutations, frameshift deletions and insertions, and in-frame deletions.
*FBXW7*	4q31.3	F-box/WD repeat-containing protein 7. F-box and WD repeat domain containing 7, E3 ubiquitin protein ligase (FBXW7) is a gene that encodes a member of the F-box protein family.	Missense, nonsense, silent, and frameshift insertions and deletions.
*ATM*	11q22.3	DNA damage/repair. ATM serine/threonine kinase (ATM) is a gene that encodes a protein that is a member of the PI3/PI4-kinase family. The protein functions as a cell cycle checkpoint kinase and regulates multiple downstream effectors.	Missense mutations, nonsense mutations, silent mutations, whole gene deletions, frameshift deletions and insertions, and in-frame deletions and insertions.
*LATS2*	13q12.11	Serine/threonine-protein kinase LATS2.	The most common alterations in LATS2 are LATS2 Amplification (0.33%), LATS2 Loss (0.20%), LATS2 P479_A480dup (0.02%), LATS2 R1054* (0.02%), and LATS2 A476T (0.02%).
*CREBBP*	16p13.3	CREB-binding protein. CREB-binding protein (CREBBP) is a gene that encodes a protein that functions in transcriptional activation and is involved in the regulation of embryonic development, growth control, and homeostasis. The protein also acetylates histone proteins.	Fusions, missense mutations, nonsense mutations, silent mutations, frameshift insertions and deletions, and in-frame.
*ARID1B*	6q25.3	AT-rich interactive domain-containing protein 1B. AT rich interactive domain 1B (SWI 1-like) (ARID1B) is a gene that encodes a protein that is a component of the SWI/SNF chromatin remodeling complex. The protein functions in cell-cycle activation.	Missense mutations, nonsense mutations, silent mutations, frameshift deletions and insertions, and in-frame deletions and insertions.
*PTEN*	10q23.31	PI3K/AKT1/MTOR. PTEN (phosphatase and tensin homolog) is a gene that encodes for phosphatidylinositol 3,4,5-trisphosphate 3-phosphatase and dual-specificity protein phosphatase PTEN. This protein is a lipid/protein phosphatase that plays a role in multiple cell processes, including growth, proliferation, survival, and maintenance of genomic integrity. PTEN acts as a tumor suppressor by negatively regulating the PI3K/AKT signaling pathway.	Somatic mutations of PTEN occur in multiple malignancies. Germline mutations of PTEN lead to inherited hamartoma and Cowden syndrome.
*TET2*	4q24	Chromatin remodeling/DNA methylation. Tet methylcytosine dioxygenase 2 (*TET2*; also known as ten-eleven translocation 2) is a gene that codes for methylcytosine dioxygenase TET2, a protein involved in epigenetic regulation of myelopoeisis.	TET2 is a tumor suppressor, and so in cancer, loss of TET2 function, which can occur via *TET2* mutation, *TET2* deletion, or *IDH1* or *IDH2* mutation, can cause myeloid or lymphoid transformations. Mutations in *TET2* have been found in MDS, AML, ALL, and other hematologic malignancies.
*DNMT3A*	2p23.3	Chromatin remodeling/DNA methylation. DNMT3A (DNA (cytosine-5-)-methyltransferase 3 alpha) gene encodes the DNA (cytosine-5)-methyltransferase 3A protein, which is involved in epigenetic gene regulation.	DNMT3A is most frequently mutated in hematologic malignancies, but it has also been observed in other cancers, including lung cancer and MPM. The most common alterations in DNMT3A are DNMT3A Mutation (2.95%), DNMT3A R882H (0.41%), DNMT3A Nonsense (0.38%), DNMT3A R882C (0.21%), and DNMT3A Loss (0.08%).
*KMT2A*	11q23.3	MLL cleavage product C180. Lysine (K)-specific methyltransferase 2A (*KMT2A*; also known as MLL) is a gene that encodes a protein that functions as a transcriptional coactivator. The protein is involved in cellular processes including the regulation of gene expression and hematopoiesis.	Fusions, rearrangements, missense mutations, nonsense mutations, silent mutations, frameshift insertions and deletions, and in-frame deletions.
*FAT1*	4q35.2	Protocadherin Fat 1, nuclear form. FAT atypical cadherin 1 (FAT1) is a gene that encodes a tumor suppressor protein that controls cell proliferation.	Missense mutations, synonymous mutations, nonsense mutations, frameshift deletions, and frameshift insertions.
*MTAP*	9p21.3	S-methyl-5’-thioadenosine phosphorylase.	The most common alterations in MTAP are MTAP Loss (4.92%), MTAP Amplification (1.28%), MTAP-RAF1 Fusion (0.04%), MTAP A191fs (0.05%), and MTAP A76V (0.04%).
*EP300*	22q13.2	Histone acetyltransferase p300. E1A binding protein p300 (EP300) is a gene that encodes a protein that functions in transcriptional regulation by histone acetylation.	Fusions, missense mutations, nonsense mutations, silent mutations, frameshift insertions and deletions, and in-frame insertions and deletions.
*PTCH1*	9q22.32	Hedgehog signaling. Patched 1 (PTCH1) is a gene that encodes a protein that belongs to the patched gene family. The protein functions as a receptor protein for sonic hedgehog, desert hedgehog, and indian hedgehog proteins.	Fusions, missense mutations, nonsense mutations, silent mutations, whole gene deletions, frameshift deletions and insertions, and in-frame deletions and insertions.
*LATS1*	6q25.1	Serine/threonine-protein kinase LATS1.	The most common alterations in LATS1 are LATS1 Amplification (0.16%), LATS1 Loss (0.14%), LATS1 R995C (0.03%), LATS1 R670W (0.02%), and LATS1 R737* (0.03%).
*RECQL4*	8q24.3	ATP-dependent DNA helicase Q4. RecQ helicase-like 4 (RECQL4) is a gene that encodes a DNA helicase that is predominantly expressed in thymus and testis.	Missense mutations, synonymous mutations, nonsense mutations, and frameshift deletions.
*ROS1*	6q22.1	Kinase fusions, receptor tyrosine kinase/growth factor signaling. ROS1 (ROS proto-oncogene 1, receptor tyrosine kinase) is a gene that encodes the proto-oncogene tyrosine-protein kinase ROS protein, a receptor tyrosine kinase (RTK) of the insulin receptor family. OS1 fusions have been described in glioblastoma and colangiocarcinoma. ROS1 fusions containing an intact tyrosine kinase domain possess oncogenic activity. Signaling downstream of ROS1 fusions results in activation of cellular pathways known to be involved in cell growth and cell proliferation.	The most common alterations in ROS1 are ROS1 Mutation (3.56%), ROS1 Fusion (0.29%), ROS1 Amplification (0.17%), ROS1 Loss (0.12%), and ROS1-CD74 Fusion (0.05%).
*WT1*	11p13	Beta-Catenin/WNT signaling. Wilms tumor 1 (WT1) is a gene that encodes a transcription factor that contains four zinc finger motifs and a proline/glutamine-rich DNA binding region at opposite termini.	Fusions, missense mutations, nonsense mutations, silent mutations and frameshift deletions.
*BCOR*	Xp11.4	BCL-6 corepressor. BCL6 corepressor (*BCOR*) is a gene that encodes the BCL-6 corepressor protein. BCOR is a member of the ankyrin repeat domain containing gene family. The corepressor expressed by BCOR binds to BCL6, a DNA-binding protein that acts as a transcription repressor for genes involved in regulation of B cells, a type of immune cell. *BCOR* mutations have been observed in myelodysplastic syndromes, endometrial cancer, and other cancers.	The most common alterations in BCOR are BCOR Mutation (3.39%), BCOR Frameshift (0.63%), BCOR Nonsense (0.44%), BCOR N1459S (0.34%), and BCOR Loss (0.22%).
*ARID2*	12q12	AT-rich interactive domain-containing protein 2. AT-rich interactive domain 2 (ARID; RFX-like) (ARID2) is a gene that encodes a protein that functions in a chromatin remodeling complex to promote gene transcription	Missense mutations, nonsense mutations, silent mutations, frameshift insertions and deletions, and in-frame insertions and deletions.
*ASXL1*	20q11.21	Chromatin remodeling/DNA methylation. Additional sex combs like transcriptional regulator 1 (official symbol *ASXL1*) is a gene that encodes the putative Polycomb group protein ASXL1. Normal ASXL1 plays a role in embryonic development.	The most common alterations in ASXL1 are ASXL1 Mutation (2.62%), ASXL1 Nonsense (0.65%), ASXL1 Amplification (0.67%), ASXL1 R693* (0.10%), and ASXL1 Y591* (0.07%).

**Table 2 ijms-24-03496-t002:** Known MPM associated genetic biomarkers.

	Location	Pathway	Variant Types
WT-1	11p13	Beta-Catenin/WNT signaling	Expression
PMS2 Deficient Expression	7p22.1		Deficient Expression
MSLN Overexpression	16p13.3		Overexpression
MSLN Expression	16p13.3		Expression
MSH6 Deficient Expression	2p16.3	Chromatin remodeling/DNA methylation	Deficient Expression
MSH2 Deficient Expression	2p21-p16.3	Chromatin remodeling/DNA methylation	Deficient Expression
MLH1 Deficient Expression	3p22.2	DNA damage/repair	Deficient Expression
HLA-A*02:01 Positive			
Deficient DNA Mismatch Repair (dMMR)		Predictive biomarker for use of nivolumab, pembrolizumab, dostarlimab, fluorouracil, and ipilimumab in patients	
BRCA2 Mutation	13q13.1	DNA damage/repair	Missense mutations, nonsense mutations, silent mutations, whole gene deletions, frameshift deletions and insertions, and in-frame deletions
BRCA2 Loss	13q13.1	DNA damage/repair	Loss
BRCA1	17q21.31	DNA damage/repair	Missense mutations, nonsense mutations, silent mutations, frameshift deletions and insertions, and in-frame deletions

## Data Availability

Not applicable.

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
