# Peer review of "The Genes–Stemness–Secretome Interplay in Malignant Pleural Mesothelioma: Molecular Dynamics and Clinical Hints"

_ijms, 2023, doi:10.3390/ijms24043496_

Round 1

Reviewer 1 Report

This review deals with some molecular aspects of the pathogenesis of the very aggressive tumor of the mesothelium, that is malignant mesothelioma (MPM). It  analyzes the role plaid by hypoxia, cancer stem cells and extracellular vesicles in the progression of this tumor. Furthermore it envisages how these factors could be useful targets to fight MPM.

Apart from the interest that this topic undoubtedly has, reading this review is difficult. The english language should be extensively revised. Some sentences are incomplete or too complex to be clearly understood and should be improved. Some important aspects of the MPM biology are not  discussed and finally the text suffers of many orthografic errors. Examples are reported as follows:

Orthografic errors: Line 23: macrovesicles, Line 95: di?, Line 109:…in animal models,  Line 109: epithelial MPM?, Line 113: identified chromosomal deletions,  Line 193: NF-kb, Line 243: epithelioid,  activation, Line 245: strongly, Line 254: tumor, Line 278 NF2 should be called either merlin or Neurofibromin (table 1),   Line 281: ….is known to be  critical, Line 284: an, Line  286: TNF-a activity, Line 295: disease, Line: 296: related, Line 298: either…or, Line: 312: therapeutic, Line: 313: maintaining, Line 322: behavior, Line 343: MPM-EVs, Line 400: together, Line 401: intervention.

Sentences/Paragraphs which should be made more clear:

English language requires an extensive revision by a English Mother Tongue person. However the following sentences should be made more understandable. Lines 84-86; Lines 98-100; Lines 157-160, Lines 170-178; Lines 183-195; Lines 251-259; Lines 267-270; Lines 309-311.

References: missing or reported in a wrong way:

Line 69: (16-20), Line 78: ..in the presence of genetic predisposition (missing), Line 119: missing, Line 220: (76-78), Line 228: (81-87), Line 251: (102-104), Line 270: Bronte??, Line 280: 119-121, Line 284: (126). The work of   Ito et al. (2021)  should be mentioned in the paragraph starting at Line 315, (Ito, F. et al. 2021. Ferroptosis-dependent extracellular vesicles from macrophage contribute to asbestos-induced mesothelial  carcinogenesis through loading ferritin. Redox Biol 47:102174. doi:10.1016/j.redox.2021.102174). Line 390: (monaco??).

Important aspects of the MPM biology which are not discussed:

The work of Crovella et al, Celsi et al , Borelli et al should be discussed.

Borelli et al. NLRP1 and NLRP3 polymorphisms in mesothelioma patients and asbestos exposed individuals a population-based autopsy study from North East Italy Infectious Agents and Cancer volume 10, Article number: 26 (2015) 

Crovella et al.  Iron signature in asbestos-induced malignant pleural mesothelioma: A population-based autopsy study. J Toxicol Environ Health A. 2016 Jan 28:1-13.

F Celsi et al. Pleural mesothelioma and lung cancer: the role of asbestos exposure and genetic variants in selected iron metabolism and inflammation genes. J Toxicol Environ Health A. 2019;82(20):1088-1102. doi: 10.1080/15287394.2019.1694612. Epub 2019 Nov 22.

Crovella et al. Biological Pathways Associated With the Development of Pulmonary Toxicities in Mesothelioma Patients Treated With Radical Hemithoracic Radiation Therapy: A Preliminary Study. Frontiers in Oncology | www.frontiersin.org 1 December 2021 | Volume 11 | Article 784081.

Crovella et al.Variant Enrichment Analysis to Explore Pathways Disruption in a Necropsy Series of Asbestos-Exposed Shipyard Workers Int. J. Mol. Sci. 2022, 23(21), 13628; https://doi.org/10.3390/ijms232113628

Some acronims are reported in a wrong way:

Line 125: EMT should be explained  here and not in Line 146, Line 251: please indicate here (OS), Line 276: explain PFS here,

Conclusions should be drawn at the end of each paragraph:

Few words of summary /conclusion should be added at the endo of each paragraph.

This review deals with some molecular aspects of the pathogenesis of the very aggressive tumor of the mesothelium, that is malignant mesothelioma (MPM). It  analyzes the role plaid by hypoxia, cancer stem cells and extracellular vesicles in the progression of this tumor. Furthermore it envisages how these factors could be useful targets to fight MPM.

Apart from the interest that this topic undoubtely has, reading this review is difficult. The english language should be revised. Some sentences are incomplete or too complex to be clearly understood and should be improved. Some important aspects of the MPM biology are not  discussed and finally the text suffers of many orthografic errors. Examples are reported as follows:

Orthografic errors: Line 23: macrovesicles, Line 95: di?, Line 109:…in animal models,  Line 109: epithelial MPM?, Line 113: identified chromosomal deletions,  Line 193: NF-kb, Line 243: epithelioid,  activation, Line 245: strongly, Line 254: tumor, Line 278 NF2 should be called either merlin or Neurofibromin (table 1),   Line 281: ….is known to be  critical, Line 284: an, Line  286: TNF-a activity, Line 295: disease, Line: 296: related, Line 298: either…or, Line: 312: therapeutic, Line: 313: maintaining, Line 322: behavior, Line 343: MPM-EVs, Line 400: together, Line 401: intervention.

Sentences/Paragraphs which should be made more clear:

English language requires an extensive revision by a English Mother Tongue person. However the following sentences should be made more understandable. Lines 84-86; Lines 98-100; Lines 157-160, Lines 170-178; Lines 183-195; Lines 251-259; Lines 267-270; Lines 309-311.

References: missing or reported in a wrong way:

Line 69: (16-20), Line 78: ..in the presence of genetic predisposition (missing), Line 119: missing, Line 220: (76-78), Line 228: (81-87), Line 251: (102-104), Line 270: Bronte??, Line 280: 119-121, Line 284: (126). The work of   Ito et al. (2021)  should be mentioned in the paragraph starting at Line 315, (Ito, F. et al. 2021. Ferroptosis-dependent extracellular vesicles from macrophage contribute to asbestos-induced mesothelial  carcinogenesis through loading ferritin. Redox Biol 47:102174. doi:10.1016/j.redox.2021.102174). Line 390: (monaco??).

Important aspects of the MPM biology which are not discussed:

The work of Crovella et al, Celsi et al , Borelli et al should be discussed.

Borelli et al. NLRP1 and NLRP3 polymorphisms in mesothelioma patients and asbestos exposed individuals a population-based autopsy study from North East Italy Infectious Agents and Cancer volume 10, Article number: 26 (2015) 

Crovella et al.  Iron signature in asbestos-induced malignant pleural mesothelioma: A population-based autopsy study. J Toxicol Environ Health A. 2016 Jan 28:1-13.

F Celsi et al. Pleural mesothelioma and lung cancer: the role of asbestos exposure and genetic variants in selected iron metabolism and inflammation genes. J Toxicol Environ Health A. 2019;82(20):1088-1102. doi: 10.1080/15287394.2019.1694612. Epub 2019 Nov 22.

Crovella et al. Biological Pathways Associated With the Development of Pulmonary Toxicities in Mesothelioma Patients Treated With Radical Hemithoracic Radiation Therapy: A Preliminary Study. Frontiers in Oncology | www.frontiersin.org 1 December 2021 | Volume 11 | Article 784081.

Crovella et al.Variant Enrichment Analysis to Explore Pathways Disruption in a Necropsy Series of Asbestos-Exposed Shipyard Workers Int. J. Mol. Sci. 2022, 23(21), 13628; https://doi.org/10.3390/ijms232113628

Some acronims are reported in a wrong way:

Line 125: EMT should be explained  here and not in Line 146, Line 251: please indicate here (OS), Line 276: explain PFS here,

Conclusions should be drawn at the end of each paragraph:

Few words of summary /conclusion should be added at the endo of each paragraph.

This review deals with some molecular aspects of the pathogenesis of the very aggressive tumor of the mesothelium, that is malignant mesothelioma (MPM). It  analyzes the role plaid by hypoxia, cancer stem cells and extracellular vesicles in the progression of this tumor. Furthermore it envisages how these factors could be useful targets to fight MPM.

Apart from the interest that this topic undoubtely has, reading this review is difficult. The english language should be revised. Some sentences are incomplete or too complex to be clearly understood and should be improved. Some important aspects of the MPM biology are not  discussed and finally the text suffers of many orthografic errors. Examples are reported as follows:

Orthografic errors: Line 23: macrovesicles, Line 95: di?, Line 109:…in animal models,  Line 109: epithelial MPM?, Line 113: identified chromosomal deletions,  Line 193: NF-kb, Line 243: epithelioid,  activation, Line 245: strongly, Line 254: tumor, Line 278 NF2 should be called either merlin or Neurofibromin (table 1),   Line 281: ….is known to be  critical, Line 284: an, Line  286: TNF-a activity, Line 295: disease, Line: 296: related, Line 298: either…or, Line: 312: therapeutic, Line: 313: maintaining, Line 322: behavior, Line 343: MPM-EVs, Line 400: together, Line 401: intervention.

Sentences/Paragraphs which should be made more clear:

English language requires an extensive revision by a English Mother Tongue person. However the following sentences should be made more understandable. Lines 84-86; Lines 98-100; Lines 157-160, Lines 170-178; Lines 183-195; Lines 251-259; Lines 267-270; Lines 309-311.

References: missing or reported in a wrong way:

Line 69: (16-20), Line 78: ..in the presence of genetic predisposition (missing), Line 119: missing, Line 220: (76-78), Line 228: (81-87), Line 251: (102-104), Line 270: Bronte??, Line 280: 119-121, Line 284: (126). The work of   Ito et al. (2021)  should be mentioned in the paragraph starting at Line 315, (Ito, F. et al. 2021. Ferroptosis-dependent extracellular vesicles from macrophage contribute to asbestos-induced mesothelial  carcinogenesis through loading ferritin. Redox Biol 47:102174. doi:10.1016/j.redox.2021.102174). Line 390: (monaco??).

Important aspects of the MPM biology which are not discussed:

The work of Crovella et al, Celsi et al , Borelli et al should be discussed.

Borelli et al. NLRP1 and NLRP3 polymorphisms in mesothelioma patients and asbestos exposed individuals a population-based autopsy study from North East Italy Infectious Agents and Cancer volume 10, Article number: 26 (2015) 

Crovella et al.  Iron signature in asbestos-induced malignant pleural mesothelioma: A population-based autopsy study. J Toxicol Environ Health A. 2016 Jan 28:1-13.

F Celsi et al. Pleural mesothelioma and lung cancer: the role of asbestos exposure and genetic variants in selected iron metabolism and inflammation genes. J Toxicol Environ Health A. 2019;82(20):1088-1102. doi: 10.1080/15287394.2019.1694612. Epub 2019 Nov 22.

Crovella et al. Biological Pathways Associated With the Development of Pulmonary Toxicities in Mesothelioma Patients Treated With Radical Hemithoracic Radiation Therapy: A Preliminary Study. Frontiers in Oncology | www.frontiersin.org 1 December 2021 | Volume 11 | Article 784081.

Crovella et al.Variant Enrichment Analysis to Explore Pathways Disruption in a Necropsy Series of Asbestos-Exposed Shipyard Workers Int. J. Mol. Sci. 2022, 23(21), 13628; https://doi.org/10.3390/ijms232113628

Some acronims are reported in a wrong way:

Line 125: EMT should be explained  here and not in Line 146, Line 251: please indicate here (OS), Line 276: explain PFS here,

Conclusions should be drawn at the end of each paragraph:

Few words of summary /conclusion should be added at the endo of each paragraph.

This review deals with some molecular aspects of the pathogenesis of the very aggressive tumor of the mesothelium, that is malignant mesothelioma (MPM). It  analyzes the role plaid by hypoxia, cancer stem cells and extracellular vesicles in the progression of this tumor. Furthermore it envisages how these factors could be useful targets to fight MPM.

Apart from the interest that this topic undoubtely has, reading this review is difficult. The english language should be revised. Some sentences are incomplete or too complex to be clearly understood and should be improved. Some important aspects of the MPM biology are not  discussed and finally the text suffers of many orthografic errors. Examples are reported as follows:

Orthografic errors: Line 23: macrovesicles, Line 95: di?, Line 109:…in animal models,  Line 109: epithelial MPM?, Line 113: identified chromosomal deletions,  Line 193: NF-kb, Line 243: epithelioid,  activation, Line 245: strongly, Line 254: tumor, Line 278 NF2 should be called either merlin or Neurofibromin (table 1),   Line 281: ….is known to be  critical, Line 284: an, Line  286: TNF-a activity, Line 295: disease, Line: 296: related, Line 298: either…or, Line: 312: therapeutic, Line: 313: maintaining, Line 322: behavior, Line 343: MPM-EVs, Line 400: together, Line 401: intervention.

Sentences/Paragraphs which should be made more clear:

English language requires an extensive revision by a English Mother Tongue person. However the following sentences should be made more understandable. Lines 84-86; Lines 98-100; Lines 157-160, Lines 170-178; Lines 183-195; Lines 251-259; Lines 267-270; Lines 309-311.

References: missing or reported in a wrong way:

Line 69: (16-20), Line 78: ..in the presence of genetic predisposition (missing), Line 119: missing, Line 220: (76-78), Line 228: (81-87), Line 251: (102-104), Line 270: Bronte??, Line 280: 119-121, Line 284: (126). The work of   Ito et al. (2021)  should be mentioned in the paragraph starting at Line 315, (Ito, F. et al. 2021. Ferroptosis-dependent extracellular vesicles from macrophage contribute to asbestos-induced mesothelial  carcinogenesis through loading ferritin. Redox Biol 47:102174. doi:10.1016/j.redox.2021.102174). Line 390: (monaco??).

Important aspects of the MPM biology which are not discussed:

The work of Crovella et al, Celsi et al , Borelli et al should be discussed.

Borelli et al. NLRP1 and NLRP3 polymorphisms in mesothelioma patients and asbestos exposed individuals a population-based autopsy study from North East Italy Infectious Agents and Cancer volume 10, Article number: 26 (2015) 

Crovella et al.  Iron signature in asbestos-induced malignant pleural mesothelioma: A population-based autopsy study. J Toxicol Environ Health A. 2016 Jan 28:1-13.

F Celsi et al. Pleural mesothelioma and lung cancer: the role of asbestos exposure and genetic variants in selected iron metabolism and inflammation genes. J Toxicol Environ Health A. 2019;82(20):1088-1102. doi: 10.1080/15287394.2019.1694612. Epub 2019 Nov 22.

Crovella et al. Biological Pathways Associated With the Development of Pulmonary Toxicities in Mesothelioma Patients Treated With Radical Hemithoracic Radiation Therapy: A Preliminary Study. Frontiers in Oncology | www.frontiersin.org 1 December 2021 | Volume 11 | Article 784081.

Crovella et al.Variant Enrichment Analysis to Explore Pathways Disruption in a Necropsy Series of Asbestos-Exposed Shipyard Workers Int. J. Mol. Sci. 2022, 23(21), 13628; https://doi.org/10.3390/ijms232113628

Some acronims are reported in a wrong way:

Line 125: EMT should be explained  here and not in Line 146, Line 251: please indicate here (OS), Line 276: explain PFS here,

Conclusions should be drawn at the end of each paragraph:

Few words of summary /conclusion should be added at the endo of each paragraph.

This review deals with some molecular aspects of the pathogenesis of the very aggressive tumor of the mesothelium, that is malignant mesothelioma (MPM). It  analyzes the role plaid by hypoxia, cancer stem cells and extracellular vesicles in the progression of this tumor. Furthermore it envisages how these factors could be useful targets to fight MPM.

Apart from the interest that this topic undoubtely has, reading this review is difficult. The english language should be revised. Some sentences are incomplete or too complex to be clearly understood and should be improved. Some important aspects of the MPM biology are not  discussed and finally the text suffers of many orthografic errors. Examples are reported as follows:

Orthografic errors: Line 23: macrovesicles, Line 95: di?, Line 109:…in animal models,  Line 109: epithelial MPM?, Line 113: identified chromosomal deletions,  Line 193: NF-kb, Line 243: epithelioid,  activation, Line 245: strongly, Line 254: tumor, Line 278 NF2 should be called either merlin or Neurofibromin (table 1),   Line 281: ….is known to be  critical, Line 284: an, Line  286: TNF-a activity, Line 295: disease, Line: 296: related, Line 298: either…or, Line: 312: therapeutic, Line: 313: maintaining, Line 322: behavior, Line 343: MPM-EVs, Line 400: together, Line 401: intervention.

Sentences/Paragraphs which should be made more clear:

English language requires an extensive revision by a English Mother Tongue person. However the following sentences should be made more understandable. Lines 84-86; Lines 98-100; Lines 157-160, Lines 170-178; Lines 183-195; Lines 251-259; Lines 267-270; Lines 309-311.

References: missing or reported in a wrong way:

Line 69: (16-20), Line 78: ..in the presence of genetic predisposition (missing), Line 119: missing, Line 220: (76-78), Line 228: (81-87), Line 251: (102-104), Line 270: Bronte??, Line 280: 119-121, Line 284: (126). The work of   Ito et al. (2021)  should be mentioned in the paragraph starting at Line 315, (Ito, F. et al. 2021. Ferroptosis-dependent extracellular vesicles from macrophage contribute to asbestos-induced mesothelial  carcinogenesis through loading ferritin. Redox Biol 47:102174. doi:10.1016/j.redox.2021.102174). Line 390: (monaco??).

Important aspects of the MPM biology which are not discussed:

The work of Crovella et al, Celsi et al , Borelli et al should be discussed.

Borelli et al. NLRP1 and NLRP3 polymorphisms in mesothelioma patients and asbestos exposed individuals a population-based autopsy study from North East Italy Infectious Agents and Cancer volume 10, Article number: 26 (2015) 

Crovella et al.  Iron signature in asbestos-induced malignant pleural mesothelioma: A population-based autopsy study. J Toxicol Environ Health A. 2016 Jan 28:1-13.

F Celsi et al. Pleural mesothelioma and lung cancer: the role of asbestos exposure and genetic variants in selected iron metabolism and inflammation genes. J Toxicol Environ Health A. 2019;82(20):1088-1102. doi: 10.1080/15287394.2019.1694612. Epub 2019 Nov 22.

Crovella et al. Biological Pathways Associated With the Development of Pulmonary Toxicities in Mesothelioma Patients Treated With Radical Hemithoracic Radiation Therapy: A Preliminary Study. Frontiers in Oncology | www.frontiersin.org 1 December 2021 | Volume 11 | Article 784081.

Crovella et al.Variant Enrichment Analysis to Explore Pathways Disruption in a Necropsy Series of Asbestos-Exposed Shipyard Workers Int. J. Mol. Sci. 2022, 23(21), 13628; https://doi.org/10.3390/ijms232113628

Some acronims are reported in a wrong way:

Line 125: EMT should be explained  here and not in Line 146, Line 251: please indicate here (OS), Line 276: explain PFS here,

Conclusions should be drawn at the end of each paragraph:

Few words of summary /conclusion should be added at the endo of each paragraph.

This review deals with some molecular aspects of the pathogenesis of the very aggressive tumor of the mesothelium, that is malignant mesothelioma (MPM). It  analyzes the role plaid by hypoxia, cancer stem cells and extracellular vesicles in the progression of this tumor. Furthermore it envisages how these factors could be useful targets to fight MPM.

Apart from the interest that this topic undoubtely has, reading this review is difficult. The english language should be revised. Some sentences are incomplete or too complex to be clearly understood and should be improved. Some important aspects of the MPM biology are not  discussed and finally the text suffers of many orthografic errors. Examples are reported as follows:

Orthografic errors: Line 23: macrovesicles, Line 95: di?, Line 109:…in animal models,  Line 109: epithelial MPM?, Line 113: identified chromosomal deletions,  Line 193: NF-kb, Line 243: epithelioid,  activation, Line 245: strongly, Line 254: tumor, Line 278 NF2 should be called either merlin or Neurofibromin (table 1),   Line 281: ….is known to be  critical, Line 284: an, Line  286: TNF-a activity, Line 295: disease, Line: 296: related, Line 298: either…or, Line: 312: therapeutic, Line: 313: maintaining, Line 322: behavior, Line 343: MPM-EVs, Line 400: together, Line 401: intervention.

Sentences/Paragraphs which should be made more clear:

English language requires an extensive revision by a English Mother Tongue person. However the following sentences should be made more understandable. Lines 84-86; Lines 98-100; Lines 157-160, Lines 170-178; Lines 183-195; Lines 251-259; Lines 267-270; Lines 309-311.

References: missing or reported in a wrong way:

Line 69: (16-20), Line 78: ..in the presence of genetic predisposition (missing), Line 119: missing, Line 220: (76-78), Line 228: (81-87), Line 251: (102-104), Line 270: Bronte??, Line 280: 119-121, Line 284: (126). The work of   Ito et al. (2021)  should be mentioned in the paragraph starting at Line 315, (Ito, F. et al. 2021. Ferroptosis-dependent extracellular vesicles from macrophage contribute to asbestos-induced mesothelial  carcinogenesis through loading ferritin. Redox Biol 47:102174. doi:10.1016/j.redox.2021.102174). Line 390: (monaco??).

Important aspects of the MPM biology which are not discussed:

The work of Crovella et al, Celsi et al , Borelli et al should be discussed.

Borelli et al. NLRP1 and NLRP3 polymorphisms in mesothelioma patients and asbestos exposed individuals a population-based autopsy study from North East Italy Infectious Agents and Cancer volume 10, Article number: 26 (2015) 

Crovella et al.  Iron signature in asbestos-induced malignant pleural mesothelioma: A population-based autopsy study. J Toxicol Environ Health A. 2016 Jan 28:1-13.

F Celsi et al. Pleural mesothelioma and lung cancer: the role of asbestos exposure and genetic variants in selected iron metabolism and inflammation genes. J Toxicol Environ Health A. 2019;82(20):1088-1102. doi: 10.1080/15287394.2019.1694612. Epub 2019 Nov 22.

Crovella et al. Biological Pathways Associated With the Development of Pulmonary Toxicities in Mesothelioma Patients Treated With Radical Hemithoracic Radiation Therapy: A Preliminary Study. Frontiers in Oncology | www.frontiersin.org 1 December 2021 | Volume 11 | Article 784081.

Crovella et al.Variant Enrichment Analysis to Explore Pathways Disruption in a Necropsy Series of Asbestos-Exposed Shipyard Workers Int. J. Mol. Sci. 2022, 23(21), 13628; https://doi.org/10.3390/ijms232113628

Some acronims are reported in a wrong way:

Line 125: EMT should be explained  here and not in Line 146, Line 251: please indicate here (OS), Line 276: explain PFS here,

Conclusions should be drawn at the end of each paragraph:

Few words of summary /conclusion should be added at the endo of each paragraph.

This review deals with some molecular aspects of the pathogenesis of the very aggressive tumor of the mesothelium, that is malignant mesothelioma (MPM). It  analyzes the role plaid by hypoxia, cancer stem cells and extracellular vesicles in the progression of this tumor. Furthermore it envisages how these factors could be useful targets to fight MPM.

Apart from the interest that this topic undoubtely has, reading this review is difficult. The english language should be revised. Some sentences are incomplete or too complex to be clearly understood and should be improved. Some important aspects of the MPM biology are not  discussed and finally the text suffers of many orthografic errors. Examples are reported as follows:

Orthografic errors: Line 23: macrovesicles, Line 95: di?, Line 109:…in animal models,  Line 109: epithelial MPM?, Line 113: identified chromosomal deletions,  Line 193: NF-kb, Line 243: epithelioid,  activation, Line 245: strongly, Line 254: tumor, Line 278 NF2 should be called either merlin or Neurofibromin (table 1),   Line 281: ….is known to be  critical, Line 284: an, Line  286: TNF-a activity, Line 295: disease, Line: 296: related, Line 298: either…or, Line: 312: therapeutic, Line: 313: maintaining, Line 322: behavior, Line 343: MPM-EVs, Line 400: together, Line 401: intervention.

Sentences/Paragraphs which should be made more clear:

English language requires an extensive revision by a English Mother Tongue person. However the following sentences should be made more understandable. Lines 84-86; Lines 98-100; Lines 157-160, Lines 170-178; Lines 183-195; Lines 251-259; Lines 267-270; Lines 309-311.

References: missing or reported in a wrong way:

Line 69: (16-20), Line 78: ..in the presence of genetic predisposition (missing), Line 119: missing, Line 220: (76-78), Line 228: (81-87), Line 251: (102-104), Line 270: Bronte??, Line 280: 119-121, Line 284: (126). The work of   Ito et al. (2021)  should be mentioned in the paragraph starting at Line 315, (Ito, F. et al. 2021. Ferroptosis-dependent extracellular vesicles from macrophage contribute to asbestos-induced mesothelial  carcinogenesis through loading ferritin. Redox Biol 47:102174. doi:10.1016/j.redox.2021.102174). Line 390: (monaco??).

Important aspects of the MPM biology which are not discussed:

The work of Crovella et al, Celsi et al , Borelli et al should be discussed.

Borelli et al. NLRP1 and NLRP3 polymorphisms in mesothelioma patients and asbestos exposed individuals a population-based autopsy study from North East Italy Infectious Agents and Cancer volume 10, Article number: 26 (2015) 

Crovella et al.  Iron signature in asbestos-induced malignant pleural mesothelioma: A population-based autopsy study. J Toxicol Environ Health A. 2016 Jan 28:1-13.

F Celsi et al. Pleural mesothelioma and lung cancer: the role of asbestos exposure and genetic variants in selected iron metabolism and inflammation genes. J Toxicol Environ Health A. 2019;82(20):1088-1102. doi: 10.1080/15287394.2019.1694612. Epub 2019 Nov 22.

Crovella et al. Biological Pathways Associated With the Development of Pulmonary Toxicities in Mesothelioma Patients Treated With Radical Hemithoracic Radiation Therapy: A Preliminary Study. Frontiers in Oncology | www.frontiersin.org 1 December 2021 | Volume 11 | Article 784081.

Crovella et al.Variant Enrichment Analysis to Explore Pathways Disruption in a Necropsy Series of Asbestos-Exposed Shipyard Workers Int. J. Mol. Sci. 2022, 23(21), 13628; https://doi.org/10.3390/ijms232113628

Some acronims are reported in a wrong way:

Line 125: EMT should be explained  here and not in Line 146, Line 251: please indicate here (OS), Line 276: explain PFS here,

Conclusions should be drawn at the end of each paragraph:

Few words of summary /conclusion should be added at the endo of each paragraph.

This review deals with some molecular aspects of the pathogenesis of the very aggressive tumor of the mesothelium, that is malignant mesothelioma (MPM). It  analyzes the role plaid by hypoxia, cancer stem cells and extracellular vesicles in the progression of this tumor. Furthermore it envisages how these factors could be useful targets to fight MPM.

Apart from the interest that this topic undoubtely has, reading this review is difficult. The english language should be revised. Some sentences are incomplete or too complex to be clearly understood and should be improved. Some important aspects of the MPM biology are not  discussed and finally the text suffers of many orthografic errors. Examples are reported as follows:

Orthografic errors: Line 23: macrovesicles, Line 95: di?, Line 109:…in animal models,  Line 109: epithelial MPM?, Line 113: identified chromosomal deletions,  Line 193: NF-kb, Line 243: epithelioid,  activation, Line 245: strongly, Line 254: tumor, Line 278 NF2 should be called either merlin or Neurofibromin (table 1),   Line 281: ….is known to be  critical, Line 284: an, Line  286: TNF-a activity, Line 295: disease, Line: 296: related, Line 298: either…or, Line: 312: therapeutic, Line: 313: maintaining, Line 322: behavior, Line 343: MPM-EVs, Line 400: together, Line 401: intervention.

Sentences/Paragraphs which should be made more clear:

English language requires an extensive revision by a English Mother Tongue person. However the following sentences should be made more understandable. Lines 84-86; Lines 98-100; Lines 157-160, Lines 170-178; Lines 183-195; Lines 251-259; Lines 267-270; Lines 309-311.

References: missing or reported in a wrong way:

Line 69: (16-20), Line 78: ..in the presence of genetic predisposition (missing), Line 119: missing, Line 220: (76-78), Line 228: (81-87), Line 251: (102-104), Line 270: Bronte??, Line 280: 119-121, Line 284: (126). The work of   Ito et al. (2021)  should be mentioned in the paragraph starting at Line 315, (Ito, F. et al. 2021. Ferroptosis-dependent extracellular vesicles from macrophage contribute to asbestos-induced mesothelial  carcinogenesis through loading ferritin. Redox Biol 47:102174. doi:10.1016/j.redox.2021.102174). Line 390: (monaco??).

Important aspects of the MPM biology which are not discussed:

The work of Crovella et al, Celsi et al , Borelli et al should be discussed.

Borelli et al. NLRP1 and NLRP3 polymorphisms in mesothelioma patients and asbestos exposed individuals a population-based autopsy study from North East Italy Infectious Agents and Cancer volume 10, Article number: 26 (2015) 

Crovella et al.  Iron signature in asbestos-induced malignant pleural mesothelioma: A population-based autopsy study. J Toxicol Environ Health A. 2016 Jan 28:1-13.

F Celsi et al. Pleural mesothelioma and lung cancer: the role of asbestos exposure and genetic variants in selected iron metabolism and inflammation genes. J Toxicol Environ Health A. 2019;82(20):1088-1102. doi: 10.1080/15287394.2019.1694612. Epub 2019 Nov 22.

Crovella et al. Biological Pathways Associated With the Development of Pulmonary Toxicities in Mesothelioma Patients Treated With Radical Hemithoracic Radiation Therapy: A Preliminary Study. Frontiers in Oncology | www.frontiersin.org 1 December 2021 | Volume 11 | Article 784081.

Crovella et al.Variant Enrichment Analysis to Explore Pathways Disruption in a Necropsy Series of Asbestos-Exposed Shipyard Workers Int. J. Mol. Sci. 2022, 23(21), 13628; https://doi.org/10.3390/ijms232113628

Some acronims are reported in a wrong way:

Line 125: EMT should be explained  here and not in Line 146, Line 251: please indicate here (OS), Line 276: explain PFS here,

Conclusions should be drawn at the end of each paragraph:

Few words of summary /conclusion should be added at the endo of each paragraph.

This review deals with some molecular aspects of the pathogenesis of the very aggressive tumor of the mesothelium, that is malignant mesothelioma (MPM). It  analyzes the role plaid by hypoxia, cancer stem cells and extracellular vesicles in the progression of this tumor. Furthermore it envisages how these factors could be useful targets to fight MPM.

Apart from the interest that this topic undoubtely has, reading this review is difficult. The english language should be revised. Some sentences are incomplete or too complex to be clearly understood and should be improved. Some important aspects of the MPM biology are not  discussed and finally the text suffers of many orthografic errors. Examples are reported as follows:

Orthografic errors: Line 23: macrovesicles, Line 95: di?, Line 109:…in animal models,  Line 109: epithelial MPM?, Line 113: identified chromosomal deletions,  Line 193: NF-kb, Line 243: epithelioid,  activation, Line 245: strongly, Line 254: tumor, Line 278 NF2 should be called either merlin or Neurofibromin (table 1),   Line 281: ….is known to be  critical, Line 284: an, Line  286: TNF-a activity, Line 295: disease, Line: 296: related, Line 298: either…or, Line: 312: therapeutic, Line: 313: maintaining, Line 322: behavior, Line 343: MPM-EVs, Line 400: together, Line 401: intervention.

Sentences/Paragraphs which should be made more clear:

English language requires an extensive revision by a English Mother Tongue person. However the following sentences should be made more understandable. Lines 84-86; Lines 98-100; Lines 157-160, Lines 170-178; Lines 183-195; Lines 251-259; Lines 267-270; Lines 309-311.

References: missing or reported in a wrong way:

Line 69: (16-20), Line 78: ..in the presence of genetic predisposition (missing), Line 119: missing, Line 220: (76-78), Line 228: (81-87), Line 251: (102-104), Line 270: Bronte??, Line 280: 119-121, Line 284: (126). The work of   Ito et al. (2021)  should be mentioned in the paragraph starting at Line 315, (Ito, F. et al. 2021. Ferroptosis-dependent extracellular vesicles from macrophage contribute to asbestos-induced mesothelial  carcinogenesis through loading ferritin. Redox Biol 47:102174. doi:10.1016/j.redox.2021.102174). Line 390: (monaco??).

Important aspects of the MPM biology which are not discussed:

The work of Crovella et al, Celsi et al , Borelli et al should be discussed.

Borelli et al. NLRP1 and NLRP3 polymorphisms in mesothelioma patients and asbestos exposed individuals a population-based autopsy study from North East Italy Infectious Agents and Cancer volume 10, Article number: 26 (2015) 

Crovella et al.  Iron signature in asbestos-induced malignant pleural mesothelioma: A population-based autopsy study. J Toxicol Environ Health A. 2016 Jan 28:1-13.

F Celsi et al. Pleural mesothelioma and lung cancer: the role of asbestos exposure and genetic variants in selected iron metabolism and inflammation genes. J Toxicol Environ Health A. 2019;82(20):1088-1102. doi: 10.1080/15287394.2019.1694612. Epub 2019 Nov 22.

Crovella et al. Biological Pathways Associated With the Development of Pulmonary Toxicities in Mesothelioma Patients Treated With Radical Hemithoracic Radiation Therapy: A Preliminary Study. Frontiers in Oncology | www.frontiersin.org 1 December 2021 | Volume 11 | Article 784081.

Crovella et al.Variant Enrichment Analysis to Explore Pathways Disruption in a Necropsy Series of Asbestos-Exposed Shipyard Workers Int. J. Mol. Sci. 2022, 23(21), 13628; https://doi.org/10.3390/ijms232113628

Some acronims are reported in a wrong way:

Line 125: EMT should be explained  here and not in Line 146, Line 251: please indicate here (OS), Line 276: explain PFS here,

Conclusions should be drawn at the end of each paragraph:

Few words of summary /conclusion should be added at the endo of each paragraph.

This review deals with some molecular aspects of the pathogenesis of the very aggressive tumor of the mesothelium, that is malignant mesothelioma (MPM). It  analyzes the role plaid by hypoxia, cancer stem cells and extracellular vesicles in the progression of this tumor. Furthermore it envisages how these factors could be useful targets to fight MPM.

Apart from the interest that this topic undoubtely has, reading this review is difficult. The english language should be revised. Some sentences are incomplete or too complex to be clearly understood and should be improved. Some important aspects of the MPM biology are not  discussed and finally the text suffers of many orthografic errors. Examples are reported as follows:

Orthografic errors: Line 23: macrovesicles, Line 95: di?, Line 109:…in animal models,  Line 109: epithelial MPM?, Line 113: identified chromosomal deletions,  Line 193: NF-kb, Line 243: epithelioid,  activation, Line 245: strongly, Line 254: tumor, Line 278 NF2 should be called either merlin or Neurofibromin (table 1),   Line 281: ….is known to be  critical, Line 284: an, Line  286: TNF-a activity, Line 295: disease, Line: 296: related, Line 298: either…or, Line: 312: therapeutic, Line: 313: maintaining, Line 322: behavior, Line 343: MPM-EVs, Line 400: together, Line 401: intervention.

Sentences/Paragraphs which should be made more clear:

English language requires an extensive revision by a English Mother Tongue person. However the following sentences should be made more understandable. Lines 84-86; Lines 98-100; Lines 157-160, Lines 170-178; Lines 183-195; Lines 251-259; Lines 267-270; Lines 309-311.

References: missing or reported in a wrong way:

Line 69: (16-20), Line 78: ..in the presence of genetic predisposition (missing), Line 119: missing, Line 220: (76-78), Line 228: (81-87), Line 251: (102-104), Line 270: Bronte??, Line 280: 119-121, Line 284: (126). The work of   Ito et al. (2021)  should be mentioned in the paragraph starting at Line 315, (Ito, F. et al. 2021. Ferroptosis-dependent extracellular vesicles from macrophage contribute to asbestos-induced mesothelial  carcinogenesis through loading ferritin. Redox Biol 47:102174. doi:10.1016/j.redox.2021.102174). Line 390: (monaco??).

Important aspects of the MPM biology which are not discussed:

The work of Crovella et al, Celsi et al , Borelli et al should be discussed.

Borelli et al. NLRP1 and NLRP3 polymorphisms in mesothelioma patients and asbestos exposed individuals a population-based autopsy study from North East Italy Infectious Agents and Cancer volume 10, Article number: 26 (2015) 

Crovella et al.  Iron signature in asbestos-induced malignant pleural mesothelioma: A population-based autopsy study. J Toxicol Environ Health A. 2016 Jan 28:1-13.

F Celsi et al. Pleural mesothelioma and lung cancer: the role of asbestos exposure and genetic variants in selected iron metabolism and inflammation genes. J Toxicol Environ Health A. 2019;82(20):1088-1102. doi: 10.1080/15287394.2019.1694612. Epub 2019 Nov 22.

Crovella et al. Biological Pathways Associated With the Development of Pulmonary Toxicities in Mesothelioma Patients Treated With Radical Hemithoracic Radiation Therapy: A Preliminary Study. Frontiers in Oncology | www.frontiersin.org 1 December 2021 | Volume 11 | Article 784081.

Crovella et al.Variant Enrichment Analysis to Explore Pathways Disruption in a Necropsy Series of Asbestos-Exposed Shipyard Workers Int. J. Mol. Sci. 2022, 23(21), 13628; https://doi.org/10.3390/ijms232113628

Some acronims are reported in a wrong way:

Line 125: EMT should be explained  here and not in Line 146, Line 251: please indicate here (OS), Line 276: explain PFS here,

Conclusions should be drawn at the end of each paragraph:

Few words of summary /conclusion should be added at the endo of each paragraph.

This review deals with some molecular aspects of the pathogenesis of the very aggressive tumor of the mesothelium, that is malignant mesothelioma (MPM). It  analyzes the role plaid by hypoxia, cancer stem cells and extracellular vesicles in the progression of this tumor. Furthermore it envisages how these factors could be useful targets to fight MPM.

Apart from the interest that this topic undoubtely has, reading this review is difficult. The english language should be revised. Some sentences are incomplete or too complex to be clearly understood and should be improved. Some important aspects of the MPM biology are not  discussed and finally the text suffers of many orthografic errors. Examples are reported as follows:

Orthografic errors: Line 23: macrovesicles, Line 95: di?, Line 109:…in animal models,  Line 109: epithelial MPM?, Line 113: identified chromosomal deletions,  Line 193: NF-kb, Line 243: epithelioid,  activation, Line 245: strongly, Line 254: tumor, Line 278 NF2 should be called either merlin or Neurofibromin (table 1),   Line 281: ….is known to be  critical, Line 284: an, Line  286: TNF-a activity, Line 295: disease, Line: 296: related, Line 298: either…or, Line: 312: therapeutic, Line: 313: maintaining, Line 322: behavior, Line 343: MPM-EVs, Line 400: together, Line 401: intervention.

Sentences/Paragraphs which should be made more clear:

English language requires an extensive revision by a English Mother Tongue person. However the following sentences should be made more understandable. Lines 84-86; Lines 98-100; Lines 157-160, Lines 170-178; Lines 183-195; Lines 251-259; Lines 267-270; Lines 309-311.

References: missing or reported in a wrong way:

Line 69: (16-20), Line 78: ..in the presence of genetic predisposition (missing), Line 119: missing, Line 220: (76-78), Line 228: (81-87), Line 251: (102-104), Line 270: Bronte??, Line 280: 119-121, Line 284: (126). The work of   Ito et al. (2021)  should be mentioned in the paragraph starting at Line 315, (Ito, F. et al. 2021. Ferroptosis-dependent extracellular vesicles from macrophage contribute to asbestos-induced mesothelial  carcinogenesis through loading ferritin. Redox Biol 47:102174. doi:10.1016/j.redox.2021.102174). Line 390: (monaco??).

Important aspects of the MPM biology which are not discussed:

The work of Crovella et al, Celsi et al , Borelli et al should be discussed.

Borelli et al. NLRP1 and NLRP3 polymorphisms in mesothelioma patients and asbestos exposed individuals a population-based autopsy study from North East Italy Infectious Agents and Cancer volume 10, Article number: 26 (2015) 

Crovella et al.  Iron signature in asbestos-induced malignant pleural mesothelioma: A population-based autopsy study. J Toxicol Environ Health A. 2016 Jan 28:1-13.

F Celsi et al. Pleural mesothelioma and lung cancer: the role of asbestos exposure and genetic variants in selected iron metabolism and inflammation genes. J Toxicol Environ Health A. 2019;82(20):1088-1102. doi: 10.1080/15287394.2019.1694612. Epub 2019 Nov 22.

Crovella et al. Biological Pathways Associated With the Development of Pulmonary Toxicities in Mesothelioma Patients Treated With Radical Hemithoracic Radiation Therapy: A Preliminary Study. Frontiers in Oncology | www.frontiersin.org 1 December 2021 | Volume 11 | Article 784081.

Crovella et al.Variant Enrichment Analysis to Explore Pathways Disruption in a Necropsy Series of Asbestos-Exposed Shipyard Workers Int. J. Mol. Sci. 2022, 23(21), 13628; https://doi.org/10.3390/ijms232113628

Some acronims are reported in a wrong way:

Line 125: EMT should be explained  here and not in Line 146, Line 251: please indicate here (OS), Line 276: explain PFS here,

Conclusions should be drawn at the end of each paragraph:

Few words of summary /conclusion should be added at the endo of each paragraph.

This review deals with some molecular aspects of the pathogenesis of the very aggressive tumor of the mesothelium, that is malignant mesothelioma (MPM). It  analyzes the role plaid by hypoxia, cancer stem cells and extracellular vesicles in the progression of this tumor. Furthermore it envisages how these factors could be useful targets to fight MPM.

Apart from the interest that this topic undoubtely has, reading this review is difficult. The english language should be revised. Some sentences are incomplete or too complex to be clearly understood and should be improved. Some important aspects of the MPM biology are not  discussed and finally the text suffers of many orthografic errors. Examples are reported as follows:

Orthografic errors: Line 23: macrovesicles, Line 95: di?, Line 109:…in animal models,  Line 109: epithelial MPM?, Line 113: identified chromosomal deletions,  Line 193: NF-kb, Line 243: epithelioid,  activation, Line 245: strongly, Line 254: tumor, Line 278 NF2 should be called either merlin or Neurofibromin (table 1),   Line 281: ….is known to be  critical, Line 284: an, Line  286: TNF-a activity, Line 295: disease, Line: 296: related, Line 298: either…or, Line: 312: therapeutic, Line: 313: maintaining, Line 322: behavior, Line 343: MPM-EVs, Line 400: together, Line 401: intervention.

Sentences/Paragraphs which should be made more clear:

English language requires an extensive revision by a English Mother Tongue person. However the following sentences should be made more understandable. Lines 84-86; Lines 98-100; Lines 157-160, Lines 170-178; Lines 183-195; Lines 251-259; Lines 267-270; Lines 309-311.

References: missing or reported in a wrong way:

Line 69: (16-20), Line 78: ..in the presence of genetic predisposition (missing), Line 119: missing, Line 220: (76-78), Line 228: (81-87), Line 251: (102-104), Line 270: Bronte??, Line 280: 119-121, Line 284: (126). The work of   Ito et al. (2021)  should be mentioned in the paragraph starting at Line 315, (Ito, F. et al. 2021. Ferroptosis-dependent extracellular vesicles from macrophage contribute to asbestos-induced mesothelial  carcinogenesis through loading ferritin. Redox Biol 47:102174. doi:10.1016/j.redox.2021.102174). Line 390: (monaco??).

Important aspects of the MPM biology which are not discussed:

The work of Crovella et al, Celsi et al , Borelli et al should be discussed.

Borelli et al. NLRP1 and NLRP3 polymorphisms in mesothelioma patients and asbestos exposed individuals a population-based autopsy study from North East Italy Infectious Agents and Cancer volume 10, Article number: 26 (2015) 

Crovella et al.  Iron signature in asbestos-induced malignant pleural mesothelioma: A population-based autopsy study. J Toxicol Environ Health A. 2016 Jan 28:1-13.

F Celsi et al. Pleural mesothelioma and lung cancer: the role of asbestos exposure and genetic variants in selected iron metabolism and inflammation genes. J Toxicol Environ Health A. 2019;82(20):1088-1102. doi: 10.1080/15287394.2019.1694612. Epub 2019 Nov 22.

Crovella et al. Biological Pathways Associated With the Development of Pulmonary Toxicities in Mesothelioma Patients Treated With Radical Hemithoracic Radiation Therapy: A Preliminary Study. Frontiers in Oncology | www.frontiersin.org 1 December 2021 | Volume 11 | Article 784081.

Crovella et al.Variant Enrichment Analysis to Explore Pathways Disruption in a Necropsy Series of Asbestos-Exposed Shipyard Workers Int. J. Mol. Sci. 2022, 23(21), 13628; https://doi.org/10.3390/ijms232113628

Some acronims are reported in a wrong way:

Line 125: EMT should be explained  here and not in Line 146, Line 251: please indicate here (OS), Line 276: explain PFS here,

Conclusions should be drawn at the end of each paragraph:

Few words of summary /conclusion should be added at the endo of each paragraph.

This review deals with some molecular aspects of the pathogenesis of the very aggressive tumor of the mesothelium, that is malignant mesothelioma (MPM). It  analyzes the role plaid by hypoxia, cancer stem cells and extracellular vesicles in the progression of this tumor. Furthermore it envisages how these factors could be useful targets to fight MPM.

Apart from the interest that this topic undoubtely has, reading this review is difficult. The english language should be revised. Some sentences are incomplete or too complex to be clearly understood and should be improved. Some important aspects of the MPM biology are not  discussed and finally the text suffers of many orthografic errors. Examples are reported as follows:

Orthografic errors: Line 23: macrovesicles, Line 95: di?, Line 109:…in animal models,  Line 109: epithelial MPM?, Line 113: identified chromosomal deletions,  Line 193: NF-kb, Line 243: epithelioid,  activation, Line 245: strongly, Line 254: tumor, Line 278 NF2 should be called either merlin or Neurofibromin (table 1),   Line 281: ….is known to be  critical, Line 284: an, Line  286: TNF-a activity, Line 295: disease, Line: 296: related, Line 298: either…or, Line: 312: therapeutic, Line: 313: maintaining, Line 322: behavior, Line 343: MPM-EVs, Line 400: together, Line 401: intervention.

Sentences/Paragraphs which should be made more clear:

English language requires an extensive revision by a English Mother Tongue person. However the following sentences should be made more understandable. Lines 84-86; Lines 98-100; Lines 157-160, Lines 170-178; Lines 183-195; Lines 251-259; Lines 267-270; Lines 309-311.

References: missing or reported in a wrong way:

Line 69: (16-20), Line 78: ..in the presence of genetic predisposition (missing), Line 119: missing, Line 220: (76-78), Line 228: (81-87), Line 251: (102-104), Line 270: Bronte??, Line 280: 119-121, Line 284: (126). The work of   Ito et al. (2021)  should be mentioned in the paragraph starting at Line 315, (Ito, F. et al. 2021. Ferroptosis-dependent extracellular vesicles from macrophage contribute to asbestos-induced mesothelial  carcinogenesis through loading ferritin. Redox Biol 47:102174. doi:10.1016/j.redox.2021.102174). Line 390: (monaco??).

Important aspects of the MPM biology which are not discussed:

The work of Crovella et al, Celsi et al , Borelli et al should be discussed.

Borelli et al. NLRP1 and NLRP3 polymorphisms in mesothelioma patients and asbestos exposed individuals a population-based autopsy study from North East Italy Infectious Agents and Cancer volume 10, Article number: 26 (2015) 

Crovella et al.  Iron signature in asbestos-induced malignant pleural mesothelioma: A population-based autopsy study. J Toxicol Environ Health A. 2016 Jan 28:1-13.

F Celsi et al. Pleural mesothelioma and lung cancer: the role of asbestos exposure and genetic variants in selected iron metabolism and inflammation genes. J Toxicol Environ Health A. 2019;82(20):1088-1102. doi: 10.1080/15287394.2019.1694612. Epub 2019 Nov 22.

Crovella et al. Biological Pathways Associated With the Development of Pulmonary Toxicities in Mesothelioma Patients Treated With Radical Hemithoracic Radiation Therapy: A Preliminary Study. Frontiers in Oncology | www.frontiersin.org 1 December 2021 | Volume 11 | Article 784081.

Crovella et al.Variant Enrichment Analysis to Explore Pathways Disruption in a Necropsy Series of Asbestos-Exposed Shipyard Workers Int. J. Mol. Sci. 2022, 23(21), 13628; https://doi.org/10.3390/ijms232113628

Some acronims are reported in a wrong way:

Line 125: EMT should be explained  here and not in Line 146, Line 251: please indicate here (OS), Line 276: explain PFS here,

Conclusions should be drawn at the end of each paragraph:

Few words of summary /conclusion should be added at the endo of each paragraph.

This review deals with some molecular aspects of the pathogenesis of the very aggressive tumor of the mesothelium, that is malignant mesothelioma (MPM). It  analyzes the role plaid by hypoxia, cancer stem cells and extracellular vesicles in the progression of this tumor. Furthermore it envisages how these factors could be useful targets to fight MPM.

Apart from the interest that this topic undoubtely has, reading this review is difficult. The english language should be revised. Some sentences are incomplete or too complex to be clearly understood and should be improved. Some important aspects of the MPM biology are not  discussed and finally the text suffers of many orthografic errors. Examples are reported as follows:

Orthografic errors: Line 23: macrovesicles, Line 95: di?, Line 109:…in animal models,  Line 109: epithelial MPM?, Line 113: identified chromosomal deletions,  Line 193: NF-kb, Line 243: epithelioid,  activation, Line 245: strongly, Line 254: tumor, Line 278 NF2 should be called either merlin or Neurofibromin (table 1),   Line 281: ….is known to be  critical, Line 284: an, Line  286: TNF-a activity, Line 295: disease, Line: 296: related, Line 298: either…or, Line: 312: therapeutic, Line: 313: maintaining, Line 322: behavior, Line 343: MPM-EVs, Line 400: together, Line 401: intervention.

Sentences/Paragraphs which should be made more clear:

English language requires an extensive revision by a English Mother Tongue person. However the following sentences should be made more understandable. Lines 84-86; Lines 98-100; Lines 157-160, Lines 170-178; Lines 183-195; Lines 251-259; Lines 267-270; Lines 309-311.

References: missing or reported in a wrong way:

Line 69: (16-20), Line 78: ..in the presence of genetic predisposition (missing), Line 119: missing, Line 220: (76-78), Line 228: (81-87), Line 251: (102-104), Line 270: Bronte??, Line 280: 119-121, Line 284: (126). The work of   Ito et al. (2021)  should be mentioned in the paragraph starting at Line 315, (Ito, F. et al. 2021. Ferroptosis-dependent extracellular vesicles from macrophage contribute to asbestos-induced mesothelial  carcinogenesis through loading ferritin. Redox Biol 47:102174. doi:10.1016/j.redox.2021.102174). Line 390: (monaco??).

Important aspects of the MPM biology which are not discussed:

The work of Crovella et al, Celsi et al , Borelli et al should be discussed.

Borelli et al. NLRP1 and NLRP3 polymorphisms in mesothelioma patients and asbestos exposed individuals a population-based autopsy study from North East Italy Infectious Agents and Cancer volume 10, Article number: 26 (2015) 

Crovella et al.  Iron signature in asbestos-induced malignant pleural mesothelioma: A population-based autopsy study. J Toxicol Environ Health A. 2016 Jan 28:1-13.

F Celsi et al. Pleural mesothelioma and lung cancer: the role of asbestos exposure and genetic variants in selected iron metabolism and inflammation genes. J Toxicol Environ Health A. 2019;82(20):1088-1102. doi: 10.1080/15287394.2019.1694612. Epub 2019 Nov 22.

Crovella et al. Biological Pathways Associated With the Development of Pulmonary Toxicities in Mesothelioma Patients Treated With Radical Hemithoracic Radiation Therapy: A Preliminary Study. Frontiers in Oncology | www.frontiersin.org 1 December 2021 | Volume 11 | Article 784081.

Crovella et al.Variant Enrichment Analysis to Explore Pathways Disruption in a Necropsy Series of Asbestos-Exposed Shipyard Workers Int. J. Mol. Sci. 2022, 23(21), 13628; https://doi.org/10.3390/ijms232113628

Some acronims are reported in a wrong way:

Line 125: EMT should be explained  here and not in Line 146, Line 251: please indicate here (OS), Line 276: explain PFS here,

Conclusions should be drawn at the end of each paragraph:

Few words of summary /conclusion should be added at the endo of each paragraph.

This review deals with some molecular aspects of the pathogenesis of the very aggressive tumor of the mesothelium, that is malignant mesothelioma (MPM). It  analyzes the role plaid by hypoxia, cancer stem cells and extracellular vesicles in the progression of this tumor. Furthermore it envisages how these factors could be useful targets to fight MPM.

Apart from the interest that this topic undoubtely has, reading this review is difficult. The english language should be revised. Some sentences are incomplete or too complex to be clearly understood and should be improved. Some important aspects of the MPM biology are not  discussed and finally the text suffers of many orthografic errors. Examples are reported as follows:

Orthografic errors: Line 23: macrovesicles, Line 95: di?, Line 109:…in animal models,  Line 109: epithelial MPM?, Line 113: identified chromosomal deletions,  Line 193: NF-kb, Line 243: epithelioid,  activation, Line 245: strongly, Line 254: tumor, Line 278 NF2 should be called either merlin or Neurofibromin (table 1),   Line 281: ….is known to be  critical, Line 284: an, Line  286: TNF-a activity, Line 295: disease, Line: 296: related, Line 298: either…or, Line: 312: therapeutic, Line: 313: maintaining, Line 322: behavior, Line 343: MPM-EVs, Line 400: together, Line 401: intervention.

Sentences/Paragraphs which should be made more clear:

English language requires an extensive revision by a English Mother Tongue person. However the following sentences should be made more understandable. Lines 84-86; Lines 98-100; Lines 157-160, Lines 170-178; Lines 183-195; Lines 251-259; Lines 267-270; Lines 309-311.

References: missing or reported in a wrong way:

Line 69: (16-20), Line 78: ..in the presence of genetic predisposition (missing), Line 119: missing, Line 220: (76-78), Line 228: (81-87), Line 251: (102-104), Line 270: Bronte??, Line 280: 119-121, Line 284: (126). The work of   Ito et al. (2021)  should be mentioned in the paragraph starting at Line 315, (Ito, F. et al. 2021. Ferroptosis-dependent extracellular vesicles from macrophage contribute to asbestos-induced mesothelial  carcinogenesis through loading ferritin. Redox Biol 47:102174. doi:10.1016/j.redox.2021.102174). Line 390: (monaco??).

Important aspects of the MPM biology which are not discussed:

The work of Crovella et al, Celsi et al , Borelli et al should be discussed.

Borelli et al. NLRP1 and NLRP3 polymorphisms in mesothelioma patients and asbestos exposed individuals a population-based autopsy study from North East Italy Infectious Agents and Cancer volume 10, Article number: 26 (2015) 

Crovella et al.  Iron signature in asbestos-induced malignant pleural mesothelioma: A population-based autopsy study. J Toxicol Environ Health A. 2016 Jan 28:1-13.

F Celsi et al. Pleural mesothelioma and lung cancer: the role of asbestos exposure and genetic variants in selected iron metabolism and inflammation genes. J Toxicol Environ Health A. 2019;82(20):1088-1102. doi: 10.1080/15287394.2019.1694612. Epub 2019 Nov 22.

Crovella et al. Biological Pathways Associated With the Development of Pulmonary Toxicities in Mesothelioma Patients Treated With Radical Hemithoracic Radiation Therapy: A Preliminary Study. Frontiers in Oncology | www.frontiersin.org 1 December 2021 | Volume 11 | Article 784081.

Crovella et al.Variant Enrichment Analysis to Explore Pathways Disruption in a Necropsy Series of Asbestos-Exposed Shipyard Workers Int. J. Mol. Sci. 2022, 23(21), 13628; https://doi.org/10.3390/ijms232113628

Some acronims are reported in a wrong way:

Line 125: EMT should be explained  here and not in Line 146, Line 251: please indicate here (OS), Line 276: explain PFS here,

Conclusions should be drawn at the end of each paragraph:

Few words of summary /conclusion should be added at the endo of each paragraph.

This review deals with some molecular aspects of the pathogenesis of the very aggressive tumor of the mesothelium, that is malignant mesothelioma (MPM). It  analyzes the role plaid by hypoxia, cancer stem cells and extracellular vesicles in the progression of this tumor. Furthermore it envisages how these factors could be useful targets to fight MPM.

Apart from the interest that this topic undoubtely has, reading this review is difficult. The english language should be revised. Some sentences are incomplete or too complex to be clearly understood and should be improved. Some important aspects of the MPM biology are not  discussed and finally the text suffers of many orthografic errors. Examples are reported as follows:

Orthografic errors: Line 23: macrovesicles, Line 95: di?, Line 109:…in animal models,  Line 109: epithelial MPM?, Line 113: identified chromosomal deletions,  Line 193: NF-kb, Line 243: epithelioid,  activation, Line 245: strongly, Line 254: tumor, Line 278 NF2 should be called either merlin or Neurofibromin (table 1),   Line 281: ….is known to be  critical, Line 284: an, Line  286: TNF-a activity, Line 295: disease, Line: 296: related, Line 298: either…or, Line: 312: therapeutic, Line: 313: maintaining, Line 322: behavior, Line 343: MPM-EVs, Line 400: together, Line 401: intervention.

Sentences/Paragraphs which should be made more clear:

English language requires an extensive revision by a English Mother Tongue person. However the following sentences should be made more understandable. Lines 84-86; Lines 98-100; Lines 157-160, Lines 170-178; Lines 183-195; Lines 251-259; Lines 267-270; Lines 309-311.

References: missing or reported in a wrong way:

Line 69: (16-20), Line 78: ..in the presence of genetic predisposition (missing), Line 119: missing, Line 220: (76-78), Line 228: (81-87), Line 251: (102-104), Line 270: Bronte??, Line 280: 119-121, Line 284: (126). The work of   Ito et al. (2021)  should be mentioned in the paragraph starting at Line 315, (Ito, F. et al. 2021. Ferroptosis-dependent extracellular vesicles from macrophage contribute to asbestos-induced mesothelial  carcinogenesis through loading ferritin. Redox Biol 47:102174. doi:10.1016/j.redox.2021.102174). Line 390: (monaco??).

Important aspects of the MPM biology which are not discussed:

The work of Crovella et al, Celsi et al , Borelli et al should be discussed.

Borelli et al. NLRP1 and NLRP3 polymorphisms in mesothelioma patients and asbestos exposed individuals a population-based autopsy study from North East Italy Infectious Agents and Cancer volume 10, Article number: 26 (2015) 

Crovella et al.  Iron signature in asbestos-induced malignant pleural mesothelioma: A population-based autopsy study. J Toxicol Environ Health A. 2016 Jan 28:1-13.

F Celsi et al. Pleural mesothelioma and lung cancer: the role of asbestos exposure and genetic variants in selected iron metabolism and inflammation genes. J Toxicol Environ Health A. 2019;82(20):1088-1102. doi: 10.1080/15287394.2019.1694612. Epub 2019 Nov 22.

Crovella et al. Biological Pathways Associated With the Development of Pulmonary Toxicities in Mesothelioma Patients Treated With Radical Hemithoracic Radiation Therapy: A Preliminary Study. Frontiers in Oncology | www.frontiersin.org 1 December 2021 | Volume 11 | Article 784081.

Crovella et al.Variant Enrichment Analysis to Explore Pathways Disruption in a Necropsy Series of Asbestos-Exposed Shipyard Workers Int. J. Mol. Sci. 2022, 23(21), 13628; https://doi.org/10.3390/ijms232113628

Some acronims are reported in a wrong way:

Line 125: EMT should be explained  here and not in Line 146, Line 251: please indicate here (OS), Line 276: explain PFS here,

Conclusions should be drawn at the end of each paragraph:

Few words of summary /conclusion should be added at the endo of each paragraph.

This review deals with some molecular aspects of the pathogenesis of the very aggressive tumor of the mesothelium, that is malignant mesothelioma (MPM). It  analyzes the role plaid by hypoxia, cancer stem cells and extracellular vesicles in the progression of this tumor. Furthermore it envisages how these factors could be useful targets to fight MPM.

Apart from the interest that this topic undoubtely has, reading this review is difficult. The english language should be revised. Some sentences are incomplete or too complex to be clearly understood and should be improved. Some important aspects of the MPM biology are not  discussed and finally the text suffers of many orthografic errors. Examples are reported as follows:

Orthografic errors: Line 23: macrovesicles, Line 95: di?, Line 109:…in animal models,  Line 109: epithelial MPM?, Line 113: identified chromosomal deletions,  Line 193: NF-kb, Line 243: epithelioid,  activation, Line 245: strongly, Line 254: tumor, Line 278 NF2 should be called either merlin or Neurofibromin (table 1),   Line 281: ….is known to be  critical, Line 284: an, Line  286: TNF-a activity, Line 295: disease, Line: 296: related, Line 298: either…or, Line: 312: therapeutic, Line: 313: maintaining, Line 322: behavior, Line 343: MPM-EVs, Line 400: together, Line 401: intervention.

Sentences/Paragraphs which should be made more clear:

English language requires an extensive revision by a English Mother Tongue person. However the following sentences should be made more understandable. Lines 84-86; Lines 98-100; Lines 157-160, Lines 170-178; Lines 183-195; Lines 251-259; Lines 267-270; Lines 309-311.

References: missing or reported in a wrong way:

Line 69: (16-20), Line 78: ..in the presence of genetic predisposition (missing), Line 119: missing, Line 220: (76-78), Line 228: (81-87), Line 251: (102-104), Line 270: Bronte??, Line 280: 119-121, Line 284: (126). The work of   Ito et al. (2021)  should be mentioned in the paragraph starting at Line 315, (Ito, F. et al. 2021. Ferroptosis-dependent extracellular vesicles from macrophage contribute to asbestos-induced mesothelial  carcinogenesis through loading ferritin. Redox Biol 47:102174. doi:10.1016/j.redox.2021.102174). Line 390: (monaco??).

Important aspects of the MPM biology which are not discussed:

The work of Crovella et al, Celsi et al , Borelli et al should be discussed.

Borelli et al. NLRP1 and NLRP3 polymorphisms in mesothelioma patients and asbestos exposed individuals a population-based autopsy study from North East Italy Infectious Agents and Cancer volume 10, Article number: 26 (2015) 

Crovella et al.  Iron signature in asbestos-induced malignant pleural mesothelioma: A population-based autopsy study. J Toxicol Environ Health A. 2016 Jan 28:1-13.

F Celsi et al. Pleural mesothelioma and lung cancer: the role of asbestos exposure and genetic variants in selected iron metabolism and inflammation genes. J Toxicol Environ Health A. 2019;82(20):1088-1102. doi: 10.1080/15287394.2019.1694612. Epub 2019 Nov 22.

Crovella et al. Biological Pathways Associated With the Development of Pulmonary Toxicities in Mesothelioma Patients Treated With Radical Hemithoracic Radiation Therapy: A Preliminary Study. Frontiers in Oncology | www.frontiersin.org 1 December 2021 | Volume 11 | Article 784081.

Crovella et al.Variant Enrichment Analysis to Explore Pathways Disruption in a Necropsy Series of Asbestos-Exposed Shipyard Workers Int. J. Mol. Sci. 2022, 23(21), 13628; https://doi.org/10.3390/ijms232113628

Some acronims are reported in a wrong way:

Line 125: EMT should be explained  here and not in Line 146, Line 251: please indicate here (OS), Line 276: explain PFS here,

Conclusions should be drawn at the end of each paragraph:

Few words of summary /conclusion should be added at the endo of each paragraph.

This review deals with some molecular aspects of the pathogenesis of the very aggressive tumor of the mesothelium, that is malignant mesothelioma (MPM). It  analyzes the role plaid by hypoxia, cancer stem cells and extracellular vesicles in the progression of this tumor. Furthermore it envisages how these factors could be useful targets to fight MPM.

Apart from the interest that this topic undoubtely has, reading this review is difficult. The english language should be revised. Some sentences are incomplete or too complex to be clearly understood and should be improved. Some important aspects of the MPM biology are not  discussed and finally the text suffers of many orthografic errors. Examples are reported as follows:

Orthografic errors: Line 23: macrovesicles, Line 95: di?, Line 109:…in animal models,  Line 109: epithelial MPM?, Line 113: identified chromosomal deletions,  Line 193: NF-kb, Line 243: epithelioid,  activation, Line 245: strongly, Line 254: tumor, Line 278 NF2 should be called either merlin or Neurofibromin (table 1),   Line 281: ….is known to be  critical, Line 284: an, Line  286: TNF-a activity, Line 295: disease, Line: 296: related, Line 298: either…or, Line: 312: therapeutic, Line: 313: maintaining, Line 322: behavior, Line 343: MPM-EVs, Line 400: together, Line 401: intervention.

Sentences/Paragraphs which should be made more clear:

English language requires an extensive revision by a English Mother Tongue person. However the following sentences should be made more understandable. Lines 84-86; Lines 98-100; Lines 157-160, Lines 170-178; Lines 183-195; Lines 251-259; Lines 267-270; Lines 309-311.

References: missing or reported in a wrong way:

Line 69: (16-20), Line 78: ..in the presence of genetic predisposition (missing), Line 119: missing, Line 220: (76-78), Line 228: (81-87), Line 251: (102-104), Line 270: Bronte??, Line 280: 119-121, Line 284: (126). The work of   Ito et al. (2021)  should be mentioned in the paragraph starting at Line 315, (Ito, F. et al. 2021. Ferroptosis-dependent extracellular vesicles from macrophage contribute to asbestos-induced mesothelial  carcinogenesis through loading ferritin. Redox Biol 47:102174. doi:10.1016/j.redox.2021.102174). Line 390: (monaco??).

Important aspects of the MPM biology which are not discussed:

The work of Crovella et al, Celsi et al , Borelli et al should be discussed.

Borelli et al. NLRP1 and NLRP3 polymorphisms in mesothelioma patients and asbestos exposed individuals a population-based autopsy study from North East Italy Infectious Agents and Cancer volume 10, Article number: 26 (2015) 

Crovella et al.  Iron signature in asbestos-induced malignant pleural mesothelioma: A population-based autopsy study. J Toxicol Environ Health A. 2016 Jan 28:1-13.

F Celsi et al. Pleural mesothelioma and lung cancer: the role of asbestos exposure and genetic variants in selected iron metabolism and inflammation genes. J Toxicol Environ Health A. 2019;82(20):1088-1102. doi: 10.1080/15287394.2019.1694612. Epub 2019 Nov 22.

Crovella et al. Biological Pathways Associated With the Development of Pulmonary Toxicities in Mesothelioma Patients Treated With Radical Hemithoracic Radiation Therapy: A Preliminary Study. Frontiers in Oncology | www.frontiersin.org 1 December 2021 | Volume 11 | Article 784081.

Crovella et al.Variant Enrichment Analysis to Explore Pathways Disruption in a Necropsy Series of Asbestos-Exposed Shipyard Workers Int. J. Mol. Sci. 2022, 23(21), 13628; https://doi.org/10.3390/ijms232113628

Some acronims are reported in a wrong way:

Line 125: EMT should be explained  here and not in Line 146, Line 251: please indicate here (OS), Line 276: explain PFS here,

Conclusions should be drawn at the end of each paragraph:

Few words of summary /conclusion should be added at the endo of each paragraph.

This review deals with some molecular aspects of the pathogenesis of the very aggressive tumor of the mesothelium, that is malignant mesothelioma (MPM). It  analyzes the role plaid by hypoxia, cancer stem cells and extracellular vesicles in the progression of this tumor. Furthermore it envisages how these factors could be useful targets to fight MPM.

Apart from the interest that this topic undoubtely has, reading this review is difficult. The english language should be revised. Some sentences are incomplete or too complex to be clearly understood and should be improved. Some important aspects of the MPM biology are not  discussed and finally the text suffers of many orthografic errors. Examples are reported as follows:

Orthografic errors: Line 23: macrovesicles, Line 95: di?, Line 109:…in animal models,  Line 109: epithelial MPM?, Line 113: identified chromosomal deletions,  Line 193: NF-kb, Line 243: epithelioid,  activation, Line 245: strongly, Line 254: tumor, Line 278 NF2 should be called either merlin or Neurofibromin (table 1),   Line 281: ….is known to be  critical, Line 284: an, Line  286: TNF-a activity, Line 295: disease, Line: 296: related, Line 298: either…or, Line: 312: therapeutic, Line: 313: maintaining, Line 322: behavior, Line 343: MPM-EVs, Line 400: together, Line 401: intervention.

Sentences/Paragraphs which should be made more clear:

English language requires an extensive revision by a English Mother Tongue person. However the following sentences should be made more understandable. Lines 84-86; Lines 98-100; Lines 157-160, Lines 170-178; Lines 183-195; Lines 251-259; Lines 267-270; Lines 309-311.

References: missing or reported in a wrong way:

Line 69: (16-20), Line 78: ..in the presence of genetic predisposition (missing), Line 119: missing, Line 220: (76-78), Line 228: (81-87), Line 251: (102-104), Line 270: Bronte??, Line 280: 119-121, Line 284: (126). The work of   Ito et al. (2021)  should be mentioned in the paragraph starting at Line 315, (Ito, F. et al. 2021. Ferroptosis-dependent extracellular vesicles from macrophage contribute to asbestos-induced mesothelial  carcinogenesis through loading ferritin. Redox Biol 47:102174. doi:10.1016/j.redox.2021.102174). Line 390: (monaco??).

Important aspects of the MPM biology which are not discussed:

The work of Crovella et al, Celsi et al , Borelli et al should be discussed.

Borelli et al. NLRP1 and NLRP3 polymorphisms in mesothelioma patients and asbestos exposed individuals a population-based autopsy study from North East Italy Infectious Agents and Cancer volume 10, Article number: 26 (2015) 

Crovella et al.  Iron signature in asbestos-induced malignant pleural mesothelioma: A population-based autopsy study. J Toxicol Environ Health A. 2016 Jan 28:1-13.

F Celsi et al. Pleural mesothelioma and lung cancer: the role of asbestos exposure and genetic variants in selected iron metabolism and inflammation genes. J Toxicol Environ Health A. 2019;82(20):1088-1102. doi: 10.1080/15287394.2019.1694612. Epub 2019 Nov 22.

Crovella et al. Biological Pathways Associated With the Development of Pulmonary Toxicities in Mesothelioma Patients Treated With Radical Hemithoracic Radiation Therapy: A Preliminary Study. Frontiers in Oncology | www.frontiersin.org 1 December 2021 | Volume 11 | Article 784081.

Crovella et al.Variant Enrichment Analysis to Explore Pathways Disruption in a Necropsy Series of Asbestos-Exposed Shipyard Workers Int. J. Mol. Sci. 2022, 23(21), 13628; https://doi.org/10.3390/ijms232113628

Some acronims are reported in a wrong way:

Line 125: EMT should be explained  here and not in Line 146, Line 251: please indicate here (OS), Line 276: explain PFS here,

Conclusions should be drawn at the end of each paragraph:

Few words of summary /conclusion should be added at the endo of each paragraph.

This review deals with some molecular aspects of the pathogenesis of the very aggressive tumor of the mesothelium, that is malignant mesothelioma (MPM). It  analyzes the role plaid by hypoxia, cancer stem cells and extracellular vesicles in the progression of this tumor. Furthermore it envisages how these factors could be useful targets to fight MPM.

Apart from the interest that this topic undoubtely has, reading this review is difficult. The english language should be revised. Some sentences are incomplete or too complex to be clearly understood and should be improved. Some important aspects of the MPM biology are not  discussed and finally the text suffers of many orthografic errors. Examples are reported as follows:

Orthografic errors: Line 23: macrovesicles, Line 95: di?, Line 109:…in animal models,  Line 109: epithelial MPM?, Line 113: identified chromosomal deletions,  Line 193: NF-kb, Line 243: epithelioid,  activation, Line 245: strongly, Line 254: tumor, Line 278 NF2 should be called either merlin or Neurofibromin (table 1),   Line 281: ….is known to be  critical, Line 284: an, Line  286: TNF-a activity, Line 295: disease, Line: 296: related, Line 298: either…or, Line: 312: therapeutic, Line: 313: maintaining, Line 322: behavior, Line 343: MPM-EVs, Line 400: together, Line 401: intervention.

Sentences/Paragraphs which should be made more clear:

English language requires an extensive revision by a English Mother Tongue person. However the following sentences should be made more understandable. Lines 84-86; Lines 98-100; Lines 157-160, Lines 170-178; Lines 183-195; Lines 251-259; Lines 267-270; Lines 309-311.

References: missing or reported in a wrong way:

Line 69: (16-20), Line 78: ..in the presence of genetic predisposition (missing), Line 119: missing, Line 220: (76-78), Line 228: (81-87), Line 251: (102-104), Line 270: Bronte??, Line 280: 119-121, Line 284: (126). The work of   Ito et al. (2021)  should be mentioned in the paragraph starting at Line 315, (Ito, F. et al. 2021. Ferroptosis-dependent extracellular vesicles from macrophage contribute to asbestos-induced mesothelial  carcinogenesis through loading ferritin. Redox Biol 47:102174. doi:10.1016/j.redox.2021.102174). Line 390: (monaco??).

Important aspects of the MPM biology which are not discussed:

The work of Crovella et al, Celsi et al , Borelli et al should be discussed.

Borelli et al. NLRP1 and NLRP3 polymorphisms in mesothelioma patients and asbestos exposed individuals a population-based autopsy study from North East Italy Infectious Agents and Cancer volume 10, Article number: 26 (2015) 

Crovella et al.  Iron signature in asbestos-induced malignant pleural mesothelioma: A population-based autopsy study. J Toxicol Environ Health A. 2016 Jan 28:1-13.

F Celsi et al. Pleural mesothelioma and lung cancer: the role of asbestos exposure and genetic variants in selected iron metabolism and inflammation genes. J Toxicol Environ Health A. 2019;82(20):1088-1102. doi: 10.1080/15287394.2019.1694612. Epub 2019 Nov 22.

Crovella et al. Biological Pathways Associated With the Development of Pulmonary Toxicities in Mesothelioma Patients Treated With Radical Hemithoracic Radiation Therapy: A Preliminary Study. Frontiers in Oncology | www.frontiersin.org 1 December 2021 | Volume 11 | Article 784081.

Crovella et al.Variant Enrichment Analysis to Explore Pathways Disruption in a Necropsy Series of Asbestos-Exposed Shipyard Workers Int. J. Mol. Sci. 2022, 23(21), 13628; https://doi.org/10.3390/ijms232113628

Some acronims are reported in a wrong way:

Line 125: EMT should be explained  here and not in Line 146, Line 251: please indicate here (OS), Line 276: explain PFS here,

Conclusions should be drawn at the end of each paragraph:

Few words of summary /conclusion should be added at the endo of each paragraph.

This review deals with some molecular aspects of the pathogenesis of the very aggressive tumor of the mesothelium, that is malignant mesothelioma (MPM). It  analyzes the role plaid by hypoxia, cancer stem cells and extracellular vesicles in the progression of this tumor. Furthermore it envisages how these factors could be useful targets to fight MPM.

Apart from the interest that this topic undoubtely has, reading this review is difficult. The english language should be revised. Some sentences are incomplete or too complex to be clearly understood and should be improved. Some important aspects of the MPM biology are not  discussed and finally the text suffers of many orthografic errors. Examples are reported as follows:

Orthografic errors: Line 23: macrovesicles, Line 95: di?, Line 109:…in animal models,  Line 109: epithelial MPM?, Line 113: identified chromosomal deletions,  Line 193: NF-kb, Line 243: epithelioid,  activation, Line 245: strongly, Line 254: tumor, Line 278 NF2 should be called either merlin or Neurofibromin (table 1),   Line 281: ….is known to be  critical, Line 284: an, Line  286: TNF-a activity, Line 295: disease, Line: 296: related, Line 298: either…or, Line: 312: therapeutic, Line: 313: maintaining, Line 322: behavior, Line 343: MPM-EVs, Line 400: together, Line 401: intervention.

Sentences/Paragraphs which should be made more clear:

English language requires an extensive revision by a English Mother Tongue person. However the following sentences should be made more understandable. Lines 84-86; Lines 98-100; Lines 157-160, Lines 170-178; Lines 183-195; Lines 251-259; Lines 267-270; Lines 309-311.

References: missing or reported in a wrong way:

Line 69: (16-20), Line 78: ..in the presence of genetic predisposition (missing), Line 119: missing, Line 220: (76-78), Line 228: (81-87), Line 251: (102-104), Line 270: Bronte??, Line 280: 119-121, Line 284: (126). The work of   Ito et al. (2021)  should be mentioned in the paragraph starting at Line 315, (Ito, F. et al. 2021. Ferroptosis-dependent extracellular vesicles from macrophage contribute to asbestos-induced mesothelial  carcinogenesis through loading ferritin. Redox Biol 47:102174. doi:10.1016/j.redox.2021.102174). Line 390: (monaco??).

Important aspects of the MPM biology which are not discussed:

The work of Crovella et al, Celsi et al , Borelli et al should be discussed.

Borelli et al. NLRP1 and NLRP3 polymorphisms in mesothelioma patients and asbestos exposed individuals a population-based autopsy study from North East Italy Infectious Agents and Cancer volume 10, Article number: 26 (2015) 

Crovella et al.  Iron signature in asbestos-induced malignant pleural mesothelioma: A population-based autopsy study. J Toxicol Environ Health A. 2016 Jan 28:1-13.

F Celsi et al. Pleural mesothelioma and lung cancer: the role of asbestos exposure and genetic variants in selected iron metabolism and inflammation genes. J Toxicol Environ Health A. 2019;82(20):1088-1102. doi: 10.1080/15287394.2019.1694612. Epub 2019 Nov 22.

Crovella et al. Biological Pathways Associated With the Development of Pulmonary Toxicities in Mesothelioma Patients Treated With Radical Hemithoracic Radiation Therapy: A Preliminary Study. Frontiers in Oncology | www.frontiersin.org 1 December 2021 | Volume 11 | Article 784081.

Crovella et al.Variant Enrichment Analysis to Explore Pathways Disruption in a Necropsy Series of Asbestos-Exposed Shipyard Workers Int. J. Mol. Sci. 2022, 23(21), 13628; https://doi.org/10.3390/ijms232113628

Some acronims are reported in a wrong way:

Line 125: EMT should be explained  here and not in Line 146, Line 251: please indicate here (OS), Line 276: explain PFS here,

Conclusions should be drawn at the end of each paragraph:

Few words of summary /conclusion should be added at the endo of each paragraph.

This review deals with some molecular aspects of the pathogenesis of the very aggressive tumor of the mesothelium, that is malignant mesothelioma (MPM). It  analyzes the role plaid by hypoxia, cancer stem cells and extracellular vesicles in the progression of this tumor. Furthermore it envisages how these factors could be useful targets to fight MPM.

Apart from the interest that this topic undoubtely has, reading this review is difficult. The english language should be revised. Some sentences are incomplete or too complex to be clearly understood and should be improved. Some important aspects of the MPM biology are not  discussed and finally the text suffers of many orthografic errors. Examples are reported as follows:

Orthografic errors: Line 23: macrovesicles, Line 95: di?, Line 109:…in animal models,  Line 109: epithelial MPM?, Line 113: identified chromosomal deletions,  Line 193: NF-kb, Line 243: epithelioid,  activation, Line 245: strongly, Line 254: tumor, Line 278 NF2 should be called either merlin or Neurofibromin (table 1),   Line 281: ….is known to be  critical, Line 284: an, Line  286: TNF-a activity, Line 295: disease, Line: 296: related, Line 298: either…or, Line: 312: therapeutic, Line: 313: maintaining, Line 322: behavior, Line 343: MPM-EVs, Line 400: together, Line 401: intervention.

Sentences/Paragraphs which should be made more clear:

English language requires an extensive revision by a English Mother Tongue person. However the following sentences should be made more understandable. Lines 84-86; Lines 98-100; Lines 157-160, Lines 170-178; Lines 183-195; Lines 251-259; Lines 267-270; Lines 309-311.

References: missing or reported in a wrong way:

Line 69: (16-20), Line 78: ..in the presence of genetic predisposition (missing), Line 119: missing, Line 220: (76-78), Line 228: (81-87), Line 251: (102-104), Line 270: Bronte??, Line 280: 119-121, Line 284: (126). The work of   Ito et al. (2021)  should be mentioned in the paragraph starting at Line 315, (Ito, F. et al. 2021. Ferroptosis-dependent extracellular vesicles from macrophage contribute to asbestos-induced mesothelial  carcinogenesis through loading ferritin. Redox Biol 47:102174. doi:10.1016/j.redox.2021.102174). Line 390: (monaco??).

Important aspects of the MPM biology which are not discussed:

The work of Crovella et al, Celsi et al , Borelli et al should be discussed.

Borelli et al. NLRP1 and NLRP3 polymorphisms in mesothelioma patients and asbestos exposed individuals a population-based autopsy study from North East Italy Infectious Agents and Cancer volume 10, Article number: 26 (2015) 

Crovella et al.  Iron signature in asbestos-induced malignant pleural mesothelioma: A population-based autopsy study. J Toxicol Environ Health A. 2016 Jan 28:1-13.

F Celsi et al. Pleural mesothelioma and lung cancer: the role of asbestos exposure and genetic variants in selected iron metabolism and inflammation genes. J Toxicol Environ Health A. 2019;82(20):1088-1102. doi: 10.1080/15287394.2019.1694612. Epub 2019 Nov 22.

Crovella et al. Biological Pathways Associated With the Development of Pulmonary Toxicities in Mesothelioma Patients Treated With Radical Hemithoracic Radiation Therapy: A Preliminary Study. Frontiers in Oncology | www.frontiersin.org 1 December 2021 | Volume 11 | Article 784081.

Crovella et al.Variant Enrichment Analysis to Explore Pathways Disruption in a Necropsy Series of Asbestos-Exposed Shipyard Workers Int. J. Mol. Sci. 2022, 23(21), 13628; https://doi.org/10.3390/ijms232113628

Some acronims are reported in a wrong way:

Line 125: EMT should be explained  here and not in Line 146, Line 251: please indicate here (OS), Line 276: explain PFS here,

Conclusions should be drawn at the end of each paragraph:

Few words of summary /conclusion should be added at the endo of each paragraph.

This review deals with some molecular aspects of the pathogenesis of the very aggressive tumor of the mesothelium, that is malignant mesothelioma (MPM). It  analyzes the role plaid by hypoxia, cancer stem cells and extracellular vesicles in the progression of this tumor. Furthermore it envisages how these factors could be useful targets to fight MPM.

Apart from the interest that this topic undoubtely has, reading this review is difficult. The english language should be revised. Some sentences are incomplete or too complex to be clearly understood and should be improved. Some important aspects of the MPM biology are not  discussed and finally the text suffers of many orthografic errors. Examples are reported as follows:

Orthografic errors: Line 23: macrovesicles, Line 95: di?, Line 109:…in animal models,  Line 109: epithelial MPM?, Line 113: identified chromosomal deletions,  Line 193: NF-kb, Line 243: epithelioid,  activation, Line 245: strongly, Line 254: tumor, Line 278 NF2 should be called either merlin or Neurofibromin (table 1),   Line 281: ….is known to be  critical, Line 284: an, Line  286: TNF-a activity, Line 295: disease, Line: 296: related, Line 298: either…or, Line: 312: therapeutic, Line: 313: maintaining, Line 322: behavior, Line 343: MPM-EVs, Line 400: together, Line 401: intervention.

Sentences/Paragraphs which should be made more clear:

English language requires an extensive revision by a English Mother Tongue person. However the following sentences should be made more understandable. Lines 84-86; Lines 98-100; Lines 157-160, Lines 170-178; Lines 183-195; Lines 251-259; Lines 267-270; Lines 309-311.

References: missing or reported in a wrong way:

Line 69: (16-20), Line 78: ..in the presence of genetic predisposition (missing), Line 119: missing, Line 220: (76-78), Line 228: (81-87), Line 251: (102-104), Line 270: Bronte??, Line 280: 119-121, Line 284: (126). The work of   Ito et al. (2021)  should be mentioned in the paragraph starting at Line 315, (Ito, F. et al. 2021. Ferroptosis-dependent extracellular vesicles from macrophage contribute to asbestos-induced mesothelial  carcinogenesis through loading ferritin. Redox Biol 47:102174. doi:10.1016/j.redox.2021.102174). Line 390: (monaco??).

Important aspects of the MPM biology which are not discussed:

The work of Crovella et al, Celsi et al , Borelli et al should be discussed.

Borelli et al. NLRP1 and NLRP3 polymorphisms in mesothelioma patients and asbestos exposed individuals a population-based autopsy study from North East Italy Infectious Agents and Cancer volume 10, Article number: 26 (2015) 

Crovella et al.  Iron signature in asbestos-induced malignant pleural mesothelioma: A population-based autopsy study. J Toxicol Environ Health A. 2016 Jan 28:1-13.

F Celsi et al. Pleural mesothelioma and lung cancer: the role of asbestos exposure and genetic variants in selected iron metabolism and inflammation genes. J Toxicol Environ Health A. 2019;82(20):1088-1102. doi: 10.1080/15287394.2019.1694612. Epub 2019 Nov 22.

Crovella et al. Biological Pathways Associated With the Development of Pulmonary Toxicities in Mesothelioma Patients Treated With Radical Hemithoracic Radiation Therapy: A Preliminary Study. Frontiers in Oncology | www.frontiersin.org 1 December 2021 | Volume 11 | Article 784081.

Crovella et al.Variant Enrichment Analysis to Explore Pathways Disruption in a Necropsy Series of Asbestos-Exposed Shipyard Workers Int. J. Mol. Sci. 2022, 23(21), 13628; https://doi.org/10.3390/ijms232113628

Some acronims are reported in a wrong way:

Line 125: EMT should be explained  here and not in Line 146, Line 251: please indicate here (OS), Line 276: explain PFS here,

Conclusions should be drawn at the end of each paragraph:

Few words of summary /conclusion should be added at the endo of each paragraph.

This review deals with some molecular aspects of the pathogenesis of the very aggressive tumor of the mesothelium, that is malignant mesothelioma (MPM). It  analyzes the role plaid by hypoxia, cancer stem cells and extracellular vesicles in the progression of this tumor. Furthermore it envisages how these factors could be useful targets to fight MPM.

Apart from the interest that this topic undoubtely has, reading this review is difficult. The english language should be revised. Some sentences are incomplete or too complex to be clearly understood and should be improved. Some important aspects of the MPM biology are not  discussed and finally the text suffers of many orthografic errors. Examples are reported as follows:

Orthografic errors: Line 23: macrovesicles, Line 95: di?, Line 109:…in animal models,  Line 109: epithelial MPM?, Line 113: identified chromosomal deletions,  Line 193: NF-kb, Line 243: epithelioid,  activation, Line 245: strongly, Line 254: tumor, Line 278 NF2 should be called either merlin or Neurofibromin (table 1),   Line 281: ….is known to be  critical, Line 284: an, Line  286: TNF-a activity, Line 295: disease, Line: 296: related, Line 298: either…or, Line: 312: therapeutic, Line: 313: maintaining, Line 322: behavior, Line 343: MPM-EVs, Line 400: together, Line 401: intervention.

Sentences/Paragraphs which should be made more clear:

English language requires an extensive revision by a English Mother Tongue person. However the following sentences should be made more understandable. Lines 84-86; Lines 98-100; Lines 157-160, Lines 170-178; Lines 183-195; Lines 251-259; Lines 267-270; Lines 309-311.

References: missing or reported in a wrong way:

Line 69: (16-20), Line 78: ..in the presence of genetic predisposition (missing), Line 119: missing, Line 220: (76-78), Line 228: (81-87), Line 251: (102-104), Line 270: Bronte??, Line 280: 119-121, Line 284: (126). The work of   Ito et al. (2021)  should be mentioned in the paragraph starting at Line 315, (Ito, F. et al. 2021. Ferroptosis-dependent extracellular vesicles from macrophage contribute to asbestos-induced mesothelial  carcinogenesis through loading ferritin. Redox Biol 47:102174. doi:10.1016/j.redox.2021.102174). Line 390: (monaco??).

Important aspects of the MPM biology which are not discussed:

The work of Crovella et al, Celsi et al , Borelli et al should be discussed.

Borelli et al. NLRP1 and NLRP3 polymorphisms in mesothelioma patients and asbestos exposed individuals a population-based autopsy study from North East Italy Infectious Agents and Cancer volume 10, Article number: 26 (2015) 

Crovella et al.  Iron signature in asbestos-induced malignant pleural mesothelioma: A population-based autopsy study. J Toxicol Environ Health A. 2016 Jan 28:1-13.

F Celsi et al. Pleural mesothelioma and lung cancer: the role of asbestos exposure and genetic variants in selected iron metabolism and inflammation genes. J Toxicol Environ Health A. 2019;82(20):1088-1102. doi: 10.1080/15287394.2019.1694612. Epub 2019 Nov 22.

Crovella et al. Biological Pathways Associated With the Development of Pulmonary Toxicities in Mesothelioma Patients Treated With Radical Hemithoracic Radiation Therapy: A Preliminary Study. Frontiers in Oncology | www.frontiersin.org 1 December 2021 | Volume 11 | Article 784081.

Crovella et al.Variant Enrichment Analysis to Explore Pathways Disruption in a Necropsy Series of Asbestos-Exposed Shipyard Workers Int. J. Mol. Sci. 2022, 23(21), 13628; https://doi.org/10.3390/ijms232113628

Some acronims are reported in a wrong way:

Line 125: EMT should be explained  here and not in Line 146, Line 251: please indicate here (OS), Line 276: explain PFS here,

Conclusions should be drawn at the end of each paragraph:

Few words of summary /conclusion should be added at the endo of each paragraph.

This review deals with some molecular aspects of the pathogenesis of the very aggressive tumor of the mesothelium, that is malignant mesothelioma (MPM). It  analyzes the role plaid by hypoxia, cancer stem cells and extracellular vesicles in the progression of this tumor. Furthermore it envisages how these factors could be useful targets to fight MPM.

Apart from the interest that this topic undoubtely has, reading this review is difficult. The english language should be revised. Some sentences are incomplete or too complex to be clearly understood and should be improved. Some important aspects of the MPM biology are not  discussed and finally the text suffers of many orthografic errors. Examples are reported as follows:

Orthografic errors: Line 23: macrovesicles, Line 95: di?, Line 109:…in animal models,  Line 109: epithelial MPM?, Line 113: identified chromosomal deletions,  Line 193: NF-kb, Line 243: epithelioid,  activation, Line 245: strongly, Line 254: tumor, Line 278 NF2 should be called either merlin or Neurofibromin (table 1),   Line 281: ….is known to be  critical, Line 284: an, Line  286: TNF-a activity, Line 295: disease, Line: 296: related, Line 298: either…or, Line: 312: therapeutic, Line: 313: maintaining, Line 322: behavior, Line 343: MPM-EVs, Line 400: together, Line 401: intervention.

Sentences/Paragraphs which should be made more clear:

English language requires an extensive revision by a English Mother Tongue person. However the following sentences should be made more understandable. Lines 84-86; Lines 98-100; Lines 157-160, Lines 170-178; Lines 183-195; Lines 251-259; Lines 267-270; Lines 309-311.

References: missing or reported in a wrong way:

Line 69: (16-20), Line 78: ..in the presence of genetic predisposition (missing), Line 119: missing, Line 220: (76-78), Line 228: (81-87), Line 251: (102-104), Line 270: Bronte??, Line 280: 119-121, Line 284: (126). The work of   Ito et al. (2021)  should be mentioned in the paragraph starting at Line 315, (Ito, F. et al. 2021. Ferroptosis-dependent extracellular vesicles from macrophage contribute to asbestos-induced mesothelial  carcinogenesis through loading ferritin. Redox Biol 47:102174. doi:10.1016/j.redox.2021.102174). Line 390: (monaco??).

Important aspects of the MPM biology which are not discussed:

The work of Crovella et al, Celsi et al , Borelli et al should be discussed.

Borelli et al. NLRP1 and NLRP3 polymorphisms in mesothelioma patients and asbestos exposed individuals a population-based autopsy study from North East Italy Infectious Agents and Cancer volume 10, Article number: 26 (2015) 

Crovella et al.  Iron signature in asbestos-induced malignant pleural mesothelioma: A population-based autopsy study. J Toxicol Environ Health A. 2016 Jan 28:1-13.

F Celsi et al. Pleural mesothelioma and lung cancer: the role of asbestos exposure and genetic variants in selected iron metabolism and inflammation genes. J Toxicol Environ Health A. 2019;82(20):1088-1102. doi: 10.1080/15287394.2019.1694612. Epub 2019 Nov 22.

Crovella et al. Biological Pathways Associated With the Development of Pulmonary Toxicities in Mesothelioma Patients Treated With Radical Hemithoracic Radiation Therapy: A Preliminary Study. Frontiers in Oncology | www.frontiersin.org 1 December 2021 | Volume 11 | Article 784081.

Crovella et al.Variant Enrichment Analysis to Explore Pathways Disruption in a Necropsy Series of Asbestos-Exposed Shipyard Workers Int. J. Mol. Sci. 2022, 23(21), 13628; https://doi.org/10.3390/ijms232113628

Some acronims are reported in a wrong way:

Line 125: EMT should be explained  here and not in Line 146, Line 251: please indicate here (OS), Line 276: explain PFS here,

Conclusions should be drawn at the end of each paragraph:

Few words of summary /conclusion should be added at the endo of each paragraph.

This review deals with some molecular aspects of the pathogenesis of the very aggressive tumor of the mesothelium, that is malignant mesothelioma (MPM). It  analyzes the role plaid by hypoxia, cancer stem cells and extracellular vesicles in the progression of this tumor. Furthermore it envisages how these factors could be useful targets to fight MPM.

Apart from the interest that this topic undoubtely has, reading this review is difficult. The english language should be revised. Some sentences are incomplete or too complex to be clearly understood and should be improved. Some important aspects of the MPM biology are not  discussed and finally the text suffers of many orthografic errors. Examples are reported as follows:

Orthografic errors: Line 23: macrovesicles, Line 95: di?, Line 109:…in animal models,  Line 109: epithelial MPM?, Line 113: identified chromosomal deletions,  Line 193: NF-kb, Line 243: epithelioid,  activation, Line 245: strongly, Line 254: tumor, Line 278 NF2 should be called either merlin or Neurofibromin (table 1),   Line 281: ….is known to be  critical, Line 284: an, Line  286: TNF-a activity, Line 295: disease, Line: 296: related, Line 298: either…or, Line: 312: therapeutic, Line: 313: maintaining, Line 322: behavior, Line 343: MPM-EVs, Line 400: together, Line 401: intervention.

Sentences/Paragraphs which should be made more clear:

English language requires an extensive revision by a English Mother Tongue person. However the following sentences should be made more understandable. Lines 84-86; Lines 98-100; Lines 157-160, Lines 170-178; Lines 183-195; Lines 251-259; Lines 267-270; Lines 309-311.

References: missing or reported in a wrong way:

Line 69: (16-20), Line 78: ..in the presence of genetic predisposition (missing), Line 119: missing, Line 220: (76-78), Line 228: (81-87), Line 251: (102-104), Line 270: Bronte??, Line 280: 119-121, Line 284: (126). The work of   Ito et al. (2021)  should be mentioned in the paragraph starting at Line 315, (Ito, F. et al. 2021. Ferroptosis-dependent extracellular vesicles from macrophage contribute to asbestos-induced mesothelial  carcinogenesis through loading ferritin. Redox Biol 47:102174. doi:10.1016/j.redox.2021.102174). Line 390: (monaco??).

Important aspects of the MPM biology which are not discussed:

The work of Crovella et al, Celsi et al , Borelli et al should be discussed.

Borelli et al. NLRP1 and NLRP3 polymorphisms in mesothelioma patients and asbestos exposed individuals a population-based autopsy study from North East Italy Infectious Agents and Cancer volume 10, Article number: 26 (2015) 

Crovella et al.  Iron signature in asbestos-induced malignant pleural mesothelioma: A population-based autopsy study. J Toxicol Environ Health A. 2016 Jan 28:1-13.

F Celsi et al. Pleural mesothelioma and lung cancer: the role of asbestos exposure and genetic variants in selected iron metabolism and inflammation genes. J Toxicol Environ Health A. 2019;82(20):1088-1102. doi: 10.1080/15287394.2019.1694612. Epub 2019 Nov 22.

Crovella et al. Biological Pathways Associated With the Development of Pulmonary Toxicities in Mesothelioma Patients Treated With Radical Hemithoracic Radiation Therapy: A Preliminary Study. Frontiers in Oncology | www.frontiersin.org 1 December 2021 | Volume 11 | Article 784081.

Crovella et al.Variant Enrichment Analysis to Explore Pathways Disruption in a Necropsy Series of Asbestos-Exposed Shipyard Workers Int. J. Mol. Sci. 2022, 23(21), 13628; https://doi.org/10.3390/ijms232113628

Some acronims are reported in a wrong way:

Line 125: EMT should be explained  here and not in Line 146, Line 251: please indicate here (OS), Line 276: explain PFS here,

Conclusions should be drawn at the end of each paragraph:

Few words of summary /conclusion should be added at the endo of each paragraph.

This review deals with some molecular aspects of the pathogenesis of the very aggressive tumor of the mesothelium, that is malignant mesothelioma (MPM). It  analyzes the role plaid by hypoxia, cancer stem cells and extracellular vesicles in the progression of this tumor. Furthermore it envisages how these factors could be useful targets to fight MPM.

Apart from the interest that this topic undoubtely has, reading this review is difficult. The english language should be revised. Some sentences are incomplete or too complex to be clearly understood and should be improved. Some important aspects of the MPM biology are not  discussed and finally the text suffers of many orthografic errors. Examples are reported as follows:

Orthografic errors: Line 23: macrovesicles, Line 95: di?, Line 109:…in animal models,  Line 109: epithelial MPM?, Line 113: identified chromosomal deletions,  Line 193: NF-kb, Line 243: epithelioid,  activation, Line 245: strongly, Line 254: tumor, Line 278 NF2 should be called either merlin or Neurofibromin (table 1),   Line 281: ….is known to be  critical, Line 284: an, Line  286: TNF-a activity, Line 295: disease, Line: 296: related, Line 298: either…or, Line: 312: therapeutic, Line: 313: maintaining, Line 322: behavior, Line 343: MPM-EVs, Line 400: together, Line 401: intervention.

Sentences/Paragraphs which should be made more clear:

English language requires an extensive revision by a English Mother Tongue person. However the following sentences should be made more understandable. Lines 84-86; Lines 98-100; Lines 157-160, Lines 170-178; Lines 183-195; Lines 251-259; Lines 267-270; Lines 309-311.

References: missing or reported in a wrong way:

Line 69: (16-20), Line 78: ..in the presence of genetic predisposition (missing), Line 119: missing, Line 220: (76-78), Line 228: (81-87), Line 251: (102-104), Line 270: Bronte??, Line 280: 119-121, Line 284: (126). The work of   Ito et al. (2021)  should be mentioned in the paragraph starting at Line 315, (Ito, F. et al. 2021. Ferroptosis-dependent extracellular vesicles from macrophage contribute to asbestos-induced mesothelial  carcinogenesis through loading ferritin. Redox Biol 47:102174. doi:10.1016/j.redox.2021.102174). Line 390: (monaco??).

Important aspects of the MPM biology which are not discussed:

The work of Crovella et al, Celsi et al , Borelli et al should be discussed.

Borelli et al. NLRP1 and NLRP3 polymorphisms in mesothelioma patients and asbestos exposed individuals a population-based autopsy study from North East Italy Infectious Agents and Cancer volume 10, Article number: 26 (2015) 

Crovella et al.  Iron signature in asbestos-induced malignant pleural mesothelioma: A population-based autopsy study. J Toxicol Environ Health A. 2016 Jan 28:1-13.

F Celsi et al. Pleural mesothelioma and lung cancer: the role of asbestos exposure and genetic variants in selected iron metabolism and inflammation genes. J Toxicol Environ Health A. 2019;82(20):1088-1102. doi: 10.1080/15287394.2019.1694612. Epub 2019 Nov 22.

Crovella et al. Biological Pathways Associated With the Development of Pulmonary Toxicities in Mesothelioma Patients Treated With Radical Hemithoracic Radiation Therapy: A Preliminary Study. Frontiers in Oncology | www.frontiersin.org 1 December 2021 | Volume 11 | Article 784081.

Crovella et al.Variant Enrichment Analysis to Explore Pathways Disruption in a Necropsy Series of Asbestos-Exposed Shipyard Workers Int. J. Mol. Sci. 2022, 23(21), 13628; https://doi.org/10.3390/ijms232113628

Some acronims are reported in a wrong way:

Line 125: EMT should be explained  here and not in Line 146, Line 251: please indicate here (OS), Line 276: explain PFS here,

Conclusions should be drawn at the end of each paragraph:

Few words of summary /conclusion should be added at the endo of each paragraph.

This review deals with some molecular aspects of the pathogenesis of the very aggressive tumor of the mesothelium, that is malignant mesothelioma (MPM). It  analyzes the role plaid by hypoxia, cancer stem cells and extracellular vesicles in the progression of this tumor. Furthermore it envisages how these factors could be useful targets to fight MPM.

Apart from the interest that this topic undoubtely has, reading this review is difficult. The english language should be revised. Some sentences are incomplete or too complex to be clearly understood and should be improved. Some important aspects of the MPM biology are not  discussed and finally the text suffers of many orthografic errors. Examples are reported as follows:

Orthografic errors: Line 23: macrovesicles, Line 95: di?, Line 109:…in animal models,  Line 109: epithelial MPM?, Line 113: identified chromosomal deletions,  Line 193: NF-kb, Line 243: epithelioid,  activation, Line 245: strongly, Line 254: tumor, Line 278 NF2 should be called either merlin or Neurofibromin (table 1),   Line 281: ….is known to be  critical, Line 284: an, Line  286: TNF-a activity, Line 295: disease, Line: 296: related, Line 298: either…or, Line: 312: therapeutic, Line: 313: maintaining, Line 322: behavior, Line 343: MPM-EVs, Line 400: together, Line 401: intervention.

Sentences/Paragraphs which should be made more clear:

English language requires an extensive revision by a English Mother Tongue person. However the following sentences should be made more understandable. Lines 84-86; Lines 98-100; Lines 157-160, Lines 170-178; Lines 183-195; Lines 251-259; Lines 267-270; Lines 309-311.

References: missing or reported in a wrong way:

Line 69: (16-20), Line 78: ..in the presence of genetic predisposition (missing), Line 119: missing, Line 220: (76-78), Line 228: (81-87), Line 251: (102-104), Line 270: Bronte??, Line 280: 119-121, Line 284: (126). The work of   Ito et al. (2021)  should be mentioned in the paragraph starting at Line 315, (Ito, F. et al. 2021. Ferroptosis-dependent extracellular vesicles from macrophage contribute to asbestos-induced mesothelial  carcinogenesis through loading ferritin. Redox Biol 47:102174. doi:10.1016/j.redox.2021.102174). Line 390: (monaco??).

Important aspects of the MPM biology which are not discussed:

The work of Crovella et al, Celsi et al , Borelli et al should be discussed.

Borelli et al. NLRP1 and NLRP3 polymorphisms in mesothelioma patients and asbestos exposed individuals a population-based autopsy study from North East Italy Infectious Agents and Cancer volume 10, Article number: 26 (2015) 

Crovella et al.  Iron signature in asbestos-induced malignant pleural mesothelioma: A population-based autopsy study. J Toxicol Environ Health A. 2016 Jan 28:1-13.

F Celsi et al. Pleural mesothelioma and lung cancer: the role of asbestos exposure and genetic variants in selected iron metabolism and inflammation genes. J Toxicol Environ Health A. 2019;82(20):1088-1102. doi: 10.1080/15287394.2019.1694612. Epub 2019 Nov 22.

Crovella et al. Biological Pathways Associated With the Development of Pulmonary Toxicities in Mesothelioma Patients Treated With Radical Hemithoracic Radiation Therapy: A Preliminary Study. Frontiers in Oncology | www.frontiersin.org 1 December 2021 | Volume 11 | Article 784081.

Crovella et al.Variant Enrichment Analysis to Explore Pathways Disruption in a Necropsy Series of Asbestos-Exposed Shipyard Workers Int. J. Mol. Sci. 2022, 23(21), 13628; https://doi.org/10.3390/ijms232113628

Some acronims are reported in a wrong way:

Line 125: EMT should be explained  here and not in Line 146, Line 251: please indicate here (OS), Line 276: explain PFS here,

Conclusions should be drawn at the end of each paragraph:

Few words of summary /conclusion should be added at the endo of each paragraph.

This review deals with some molecular aspects of the pathogenesis of the very aggressive tumor of the mesothelium, that is malignant mesothelioma (MPM). It  analyzes the role plaid by hypoxia, cancer stem cells and extracellular vesicles in the progression of this tumor. Furthermore it envisages how these factors could be useful targets to fight MPM.

Apart from the interest that this topic undoubtely has, reading this review is difficult. The english language should be revised. Some sentences are incomplete or too complex to be clearly understood and should be improved. Some important aspects of the MPM biology are not  discussed and finally the text suffers of many orthografic errors. Examples are reported as follows:

Orthografic errors: Line 23: macrovesicles, Line 95: di?, Line 109:…in animal models,  Line 109: epithelial MPM?, Line 113: identified chromosomal deletions,  Line 193: NF-kb, Line 243: epithelioid,  activation, Line 245: strongly, Line 254: tumor, Line 278 NF2 should be called either merlin or Neurofibromin (table 1),   Line 281: ….is known to be  critical, Line 284: an, Line  286: TNF-a activity, Line 295: disease, Line: 296: related, Line 298: either…or, Line: 312: therapeutic, Line: 313: maintaining, Line 322: behavior, Line 343: MPM-EVs, Line 400: together, Line 401: intervention.

Sentences/Paragraphs which should be made more clear:

English language requires an extensive revision by a English Mother Tongue person. However the following sentences should be made more understandable. Lines 84-86; Lines 98-100; Lines 157-160, Lines 170-178; Lines 183-195; Lines 251-259; Lines 267-270; Lines 309-311.

References: missing or reported in a wrong way:

Line 69: (16-20), Line 78: ..in the presence of genetic predisposition (missing), Line 119: missing, Line 220: (76-78), Line 228: (81-87), Line 251: (102-104), Line 270: Bronte??, Line 280: 119-121, Line 284: (126). The work of   Ito et al. (2021)  should be mentioned in the paragraph starting at Line 315, (Ito, F. et al. 2021. Ferroptosis-dependent extracellular vesicles from macrophage contribute to asbestos-induced mesothelial  carcinogenesis through loading ferritin. Redox Biol 47:102174. doi:10.1016/j.redox.2021.102174). Line 390: (monaco??).

Important aspects of the MPM biology which are not discussed:

The work of Crovella et al, Celsi et al , Borelli et al should be discussed.

Borelli et al. NLRP1 and NLRP3 polymorphisms in mesothelioma patients and asbestos exposed individuals a population-based autopsy study from North East Italy Infectious Agents and Cancer volume 10, Article number: 26 (2015) 

Crovella et al.  Iron signature in asbestos-induced malignant pleural mesothelioma: A population-based autopsy study. J Toxicol Environ Health A. 2016 Jan 28:1-13.

F Celsi et al. Pleural mesothelioma and lung cancer: the role of asbestos exposure and genetic variants in selected iron metabolism and inflammation genes. J Toxicol Environ Health A. 2019;82(20):1088-1102. doi: 10.1080/15287394.2019.1694612. Epub 2019 Nov 22.

Crovella et al. Biological Pathways Associated With the Development of Pulmonary Toxicities in Mesothelioma Patients Treated With Radical Hemithoracic Radiation Therapy: A Preliminary Study. Frontiers in Oncology | www.frontiersin.org 1 December 2021 | Volume 11 | Article 784081.

Crovella et al.Variant Enrichment Analysis to Explore Pathways Disruption in a Necropsy Series of Asbestos-Exposed Shipyard Workers Int. J. Mol. Sci. 2022, 23(21), 13628; https://doi.org/10.3390/ijms232113628

Some acronims are reported in a wrong way:

Line 125: EMT should be explained  here and not in Line 146, Line 251: please indicate here (OS), Line 276: explain PFS here,

Conclusions should be drawn at the end of each paragraph:

Few words of summary /conclusion should be added at the endo of each paragraph.

This review deals with some molecular aspects of the pathogenesis of the very aggressive tumor of the mesothelium, that is malignant mesothelioma (MPM). It  analyzes the role plaid by hypoxia, cancer stem cells and extracellular vesicles in the progression of this tumor. Furthermore it envisages how these factors could be useful targets to fight MPM.

Apart from the interest that this topic undoubtely has, reading this review is difficult. The english language should be revised. Some sentences are incomplete or too complex to be clearly understood and should be improved. Some important aspects of the MPM biology are not  discussed and finally the text suffers of many orthografic errors. Examples are reported as follows:

Orthografic errors: Line 23: macrovesicles, Line 95: di?, Line 109:…in animal models,  Line 109: epithelial MPM?, Line 113: identified chromosomal deletions,  Line 193: NF-kb, Line 243: epithelioid,  activation, Line 245: strongly, Line 254: tumor, Line 278 NF2 should be called either merlin or Neurofibromin (table 1),   Line 281: ….is known to be  critical, Line 284: an, Line  286: TNF-a activity, Line 295: disease, Line: 296: related, Line 298: either…or, Line: 312: therapeutic, Line: 313: maintaining, Line 322: behavior, Line 343: MPM-EVs, Line 400: together, Line 401: intervention.

Sentences/Paragraphs which should be made more clear:

English language requires an extensive revision by a English Mother Tongue person. However the following sentences should be made more understandable. Lines 84-86; Lines 98-100; Lines 157-160, Lines 170-178; Lines 183-195; Lines 251-259; Lines 267-270; Lines 309-311.

References: missing or reported in a wrong way:

Line 69: (16-20), Line 78: ..in the presence of genetic predisposition (missing), Line 119: missing, Line 220: (76-78), Line 228: (81-87), Line 251: (102-104), Line 270: Bronte??, Line 280: 119-121, Line 284: (126). The work of   Ito et al. (2021)  should be mentioned in the paragraph starting at Line 315, (Ito, F. et al. 2021. Ferroptosis-dependent extracellular vesicles from macrophage contribute to asbestos-induced mesothelial  carcinogenesis through loading ferritin. Redox Biol 47:102174. doi:10.1016/j.redox.2021.102174). Line 390: (monaco??).

Important aspects of the MPM biology which are not discussed:

The work of Crovella et al, Celsi et al , Borelli et al should be discussed.

Borelli et al. NLRP1 and NLRP3 polymorphisms in mesothelioma patients and asbestos exposed individuals a population-based autopsy study from North East Italy Infectious Agents and Cancer volume 10, Article number: 26 (2015) 

Crovella et al.  Iron signature in asbestos-induced malignant pleural mesothelioma: A population-based autopsy study. J Toxicol Environ Health A. 2016 Jan 28:1-13.

F Celsi et al. Pleural mesothelioma and lung cancer: the role of asbestos exposure and genetic variants in selected iron metabolism and inflammation genes. J Toxicol Environ Health A. 2019;82(20):1088-1102. doi: 10.1080/15287394.2019.1694612. Epub 2019 Nov 22.

Crovella et al. Biological Pathways Associated With the Development of Pulmonary Toxicities in Mesothelioma Patients Treated With Radical Hemithoracic Radiation Therapy: A Preliminary Study. Frontiers in Oncology | www.frontiersin.org 1 December 2021 | Volume 11 | Article 784081.

Crovella et al.Variant Enrichment Analysis to Explore Pathways Disruption in a Necropsy Series of Asbestos-Exposed Shipyard Workers Int. J. Mol. Sci. 2022, 23(21), 13628; https://doi.org/10.3390/ijms232113628

Some acronims are reported in a wrong way:

Line 125: EMT should be explained  here and not in Line 146, Line 251: please indicate here (OS), Line 276: explain PFS here,

Conclusions should be drawn at the end of each paragraph:

Few words of summary /conclusion should be added at the endo of each paragraph.

This review deals with some molecular aspects of the pathogenesis of the very aggressive tumor of the mesothelium, that is malignant mesothelioma (MPM). It  analyzes the role plaid by hypoxia, cancer stem cells and extracellular vesicles in the progression of this tumor. Furthermore it envisages how these factors could be useful targets to fight MPM.

Apart from the interest that this topic undoubtely has, reading this review is difficult. The english language should be revised. Some sentences are incomplete or too complex to be clearly understood and should be improved. Some important aspects of the MPM biology are not  discussed and finally the text suffers of many orthografic errors. Examples are reported as follows:

Orthografic errors: Line 23: macrovesicles, Line 95: di?, Line 109:…in animal models,  Line 109: epithelial MPM?, Line 113: identified chromosomal deletions,  Line 193: NF-kb, Line 243: epithelioid,  activation, Line 245: strongly, Line 254: tumor, Line 278 NF2 should be called either merlin or Neurofibromin (table 1),   Line 281: ….is known to be  critical, Line 284: an, Line  286: TNF-a activity, Line 295: disease, Line: 296: related, Line 298: either…or, Line: 312: therapeutic, Line: 313: maintaining, Line 322: behavior, Line 343: MPM-EVs, Line 400: together, Line 401: intervention.

Sentences/Paragraphs which should be made more clear:

English language requires an extensive revision by a English Mother Tongue person. However the following sentences should be made more understandable. Lines 84-86; Lines 98-100; Lines 157-160, Lines 170-178; Lines 183-195; Lines 251-259; Lines 267-270; Lines 309-311.

References: missing or reported in a wrong way:

Line 69: (16-20), Line 78: ..in the presence of genetic predisposition (missing), Line 119: missing, Line 220: (76-78), Line 228: (81-87), Line 251: (102-104), Line 270: Bronte??, Line 280: 119-121, Line 284: (126). The work of   Ito et al. (2021)  should be mentioned in the paragraph starting at Line 315, (Ito, F. et al. 2021. Ferroptosis-dependent extracellular vesicles from macrophage contribute to asbestos-induced mesothelial  carcinogenesis through loading ferritin. Redox Biol 47:102174. doi:10.1016/j.redox.2021.102174). Line 390: (monaco??).

Important aspects of the MPM biology which are not discussed:

The work of Crovella et al, Celsi et al , Borelli et al should be discussed.

Borelli et al. NLRP1 and NLRP3 polymorphisms in mesothelioma patients and asbestos exposed individuals a population-based autopsy study from North East Italy Infectious Agents and Cancer volume 10, Article number: 26 (2015) 

Crovella et al.  Iron signature in asbestos-induced malignant pleural mesothelioma: A population-based autopsy study. J Toxicol Environ Health A. 2016 Jan 28:1-13.

F Celsi et al. Pleural mesothelioma and lung cancer: the role of asbestos exposure and genetic variants in selected iron metabolism and inflammation genes. J Toxicol Environ Health A. 2019;82(20):1088-1102. doi: 10.1080/15287394.2019.1694612. Epub 2019 Nov 22.

Crovella et al. Biological Pathways Associated With the Development of Pulmonary Toxicities in Mesothelioma Patients Treated With Radical Hemithoracic Radiation Therapy: A Preliminary Study. Frontiers in Oncology | www.frontiersin.org 1 December 2021 | Volume 11 | Article 784081.

Crovella et al.Variant Enrichment Analysis to Explore Pathways Disruption in a Necropsy Series of Asbestos-Exposed Shipyard Workers Int. J. Mol. Sci. 2022, 23(21), 13628; https://doi.org/10.3390/ijms232113628

Some acronims are reported in a wrong way:

Line 125: EMT should be explained  here and not in Line 146, Line 251: please indicate here (OS), Line 276: explain PFS here,

Conclusions should be drawn at the end of each paragraph:

Few words of summary /conclusion should be added at the endo of each paragraph.

This review deals with some molecular aspects of the pathogenesis of the very aggressive tumor of the mesothelium, that is malignant mesothelioma (MPM). It  analyzes the role plaid by hypoxia, cancer stem cells and extracellular vesicles in the progression of this tumor. Furthermore it envisages how these factors could be useful targets to fight MPM.

Apart from the interest that this topic undoubtely has, reading this review is difficult. The english language should be revised. Some sentences are incomplete or too complex to be clearly understood and should be improved. Some important aspects of the MPM biology are not  discussed and finally the text suffers of many orthografic errors. Examples are reported as follows:

Orthografic errors: Line 23: macrovesicles, Line 95: di?, Line 109:…in animal models,  Line 109: epithelial MPM?, Line 113: identified chromosomal deletions,  Line 193: NF-kb, Line 243: epithelioid,  activation, Line 245: strongly, Line 254: tumor, Line 278 NF2 should be called either merlin or Neurofibromin (table 1),   Line 281: ….is known to be  critical, Line 284: an, Line  286: TNF-a activity, Line 295: disease, Line: 296: related, Line 298: either…or, Line: 312: therapeutic, Line: 313: maintaining, Line 322: behavior, Line 343: MPM-EVs, Line 400: together, Line 401: intervention.

Sentences/Paragraphs which should be made more clear:

English language requires an extensive revision by a English Mother Tongue person. However the following sentences should be made more understandable. Lines 84-86; Lines 98-100; Lines 157-160, Lines 170-178; Lines 183-195; Lines 251-259; Lines 267-270; Lines 309-311.

References: missing or reported in a wrong way:

Line 69: (16-20), Line 78: ..in the presence of genetic predisposition (missing), Line 119: missing, Line 220: (76-78), Line 228: (81-87), Line 251: (102-104), Line 270: Bronte??, Line 280: 119-121, Line 284: (126). The work of   Ito et al. (2021)  should be mentioned in the paragraph starting at Line 315, (Ito, F. et al. 2021. Ferroptosis-dependent extracellular vesicles from macrophage contribute to asbestos-induced mesothelial  carcinogenesis through loading ferritin. Redox Biol 47:102174. doi:10.1016/j.redox.2021.102174). Line 390: (monaco??).

Important aspects of the MPM biology which are not discussed:

The work of Crovella et al, Celsi et al , Borelli et al should be discussed.

Borelli et al. NLRP1 and NLRP3 polymorphisms in mesothelioma patients and asbestos exposed individuals a population-based autopsy study from North East Italy Infectious Agents and Cancer volume 10, Article number: 26 (2015) 

Crovella et al.  Iron signature in asbestos-induced malignant pleural mesothelioma: A population-based autopsy study. J Toxicol Environ Health A. 2016 Jan 28:1-13.

F Celsi et al. Pleural mesothelioma and lung cancer: the role of asbestos exposure and genetic variants in selected iron metabolism and inflammation genes. J Toxicol Environ Health A. 2019;82(20):1088-1102. doi: 10.1080/15287394.2019.1694612. Epub 2019 Nov 22.

Crovella et al. Biological Pathways Associated With the Development of Pulmonary Toxicities in Mesothelioma Patients Treated With Radical Hemithoracic Radiation Therapy: A Preliminary Study. Frontiers in Oncology | www.frontiersin.org 1 December 2021 | Volume 11 | Article 784081.

Crovella et al.Variant Enrichment Analysis to Explore Pathways Disruption in a Necropsy Series of Asbestos-Exposed Shipyard Workers Int. J. Mol. Sci. 2022, 23(21), 13628; https://doi.org/10.3390/ijms232113628

Some acronims are reported in a wrong way:

Line 125: EMT should be explained  here and not in Line 146, Line 251: please indicate here (OS), Line 276: explain PFS here,

Conclusions should be drawn at the end of each paragraph:

Few words of summary /conclusion should be added at the endo of each paragraph.

This review deals with some molecular aspects of the pathogenesis of the very aggressive tumor of the mesothelium, that is malignant mesothelioma (MPM). It  analyzes the role plaid by hypoxia, cancer stem cells and extracellular vesicles in the progression of this tumor. Furthermore it envisages how these factors could be useful targets to fight MPM.

Apart from the interest that this topic undoubtely has, reading this review is difficult. The english language should be revised. Some sentences are incomplete or too complex to be clearly understood and should be improved. Some important aspects of the MPM biology are not  discussed and finally the text suffers of many orthografic errors. Examples are reported as follows:

Orthografic errors: Line 23: macrovesicles, Line 95: di?, Line 109:…in animal models,  Line 109: epithelial MPM?, Line 113: identified chromosomal deletions,  Line 193: NF-kb, Line 243: epithelioid,  activation, Line 245: strongly, Line 254: tumor, Line 278 NF2 should be called either merlin or Neurofibromin (table 1),   Line 281: ….is known to be  critical, Line 284: an, Line  286: TNF-a activity, Line 295: disease, Line: 296: related, Line 298: either…or, Line: 312: therapeutic, Line: 313: maintaining, Line 322: behavior, Line 343: MPM-EVs, Line 400: together, Line 401: intervention.

Sentences/Paragraphs which should be made more clear:

English language requires an extensive revision by a English Mother Tongue person. However the following sentences should be made more understandable. Lines 84-86; Lines 98-100; Lines 157-160, Lines 170-178; Lines 183-195; Lines 251-259; Lines 267-270; Lines 309-311.

References: missing or reported in a wrong way:

Line 69: (16-20), Line 78: ..in the presence of genetic predisposition (missing), Line 119: missing, Line 220: (76-78), Line 228: (81-87), Line 251: (102-104), Line 270: Bronte??, Line 280: 119-121, Line 284: (126). The work of   Ito et al. (2021)  should be mentioned in the paragraph starting at Line 315, (Ito, F. et al. 2021. Ferroptosis-dependent extracellular vesicles from macrophage contribute to asbestos-induced mesothelial  carcinogenesis through loading ferritin. Redox Biol 47:102174. doi:10.1016/j.redox.2021.102174). Line 390: (monaco??).

Important aspects of the MPM biology which are not discussed:

The work of Crovella et al, Celsi et al , Borelli et al should be discussed.

Borelli et al. NLRP1 and NLRP3 polymorphisms in mesothelioma patients and asbestos exposed individuals a population-based autopsy study from North East Italy Infectious Agents and Cancer volume 10, Article number: 26 (2015) 

Crovella et al.  Iron signature in asbestos-induced malignant pleural mesothelioma: A population-based autopsy study. J Toxicol Environ Health A. 2016 Jan 28:1-13.

F Celsi et al. Pleural mesothelioma and lung cancer: the role of asbestos exposure and genetic variants in selected iron metabolism and inflammation genes. J Toxicol Environ Health A. 2019;82(20):1088-1102. doi: 10.1080/15287394.2019.1694612. Epub 2019 Nov 22.

Crovella et al. Biological Pathways Associated With the Development of Pulmonary Toxicities in Mesothelioma Patients Treated With Radical Hemithoracic Radiation Therapy: A Preliminary Study. Frontiers in Oncology | www.frontiersin.org 1 December 2021 | Volume 11 | Article 784081.

Crovella et al.Variant Enrichment Analysis to Explore Pathways Disruption in a Necropsy Series of Asbestos-Exposed Shipyard Workers Int. J. Mol. Sci. 2022, 23(21), 13628; https://doi.org/10.3390/ijms232113628

Some acronims are reported in a wrong way:

Line 125: EMT should be explained  here and not in Line 146, Line 251: please indicate here (OS), Line 276: explain PFS here,

Conclusions should be drawn at the end of each paragraph:

Few words of summary /conclusion should be added at the endo of each paragraph.

This review deals with some molecular aspects of the pathogenesis of the very aggressive tumor of the mesothelium, that is malignant mesothelioma (MPM). It  analyzes the role plaid by hypoxia, cancer stem cells and extracellular vesicles in the progression of this tumor. Furthermore it envisages how these factors could be useful targets to fight MPM.

Apart from the interest that this topic undoubtely has, reading this review is difficult. The english language should be revised. Some sentences are incomplete or too complex to be clearly understood and should be improved. Some important aspects of the MPM biology are not  discussed and finally the text suffers of many orthografic errors. Examples are reported as follows:

Orthografic errors: Line 23: macrovesicles, Line 95: di?, Line 109:…in animal models,  Line 109: epithelial MPM?, Line 113: identified chromosomal deletions,  Line 193: NF-kb, Line 243: epithelioid,  activation, Line 245: strongly, Line 254: tumor, Line 278 NF2 should be called either merlin or Neurofibromin (table 1),   Line 281: ….is known to be  critical, Line 284: an, Line  286: TNF-a activity, Line 295: disease, Line: 296: related, Line 298: either…or, Line: 312: therapeutic, Line: 313: maintaining, Line 322: behavior, Line 343: MPM-EVs, Line 400: together, Line 401: intervention.

Sentences/Paragraphs which should be made more clear:

English language requires an extensive revision by a English Mother Tongue person. However the following sentences should be made more understandable. Lines 84-86; Lines 98-100; Lines 157-160, Lines 170-178; Lines 183-195; Lines 251-259; Lines 267-270; Lines 309-311.

References: missing or reported in a wrong way:

Line 69: (16-20), Line 78: ..in the presence of genetic predisposition (missing), Line 119: missing, Line 220: (76-78), Line 228: (81-87), Line 251: (102-104), Line 270: Bronte??, Line 280: 119-121, Line 284: (126). The work of   Ito et al. (2021)  should be mentioned in the paragraph starting at Line 315, (Ito, F. et al. 2021. Ferroptosis-dependent extracellular vesicles from macrophage contribute to asbestos-induced mesothelial  carcinogenesis through loading ferritin. Redox Biol 47:102174. doi:10.1016/j.redox.2021.102174). Line 390: (monaco??).

Important aspects of the MPM biology which are not discussed:

The work of Crovella et al, Celsi et al , Borelli et al should be discussed.

Borelli et al. NLRP1 and NLRP3 polymorphisms in mesothelioma patients and asbestos exposed individuals a population-based autopsy study from North East Italy Infectious Agents and Cancer volume 10, Article number: 26 (2015) 

Crovella et al.  Iron signature in asbestos-induced malignant pleural mesothelioma: A population-based autopsy study. J Toxicol Environ Health A. 2016 Jan 28:1-13.

F Celsi et al. Pleural mesothelioma and lung cancer: the role of asbestos exposure and genetic variants in selected iron metabolism and inflammation genes. J Toxicol Environ Health A. 2019;82(20):1088-1102. doi: 10.1080/15287394.2019.1694612. Epub 2019 Nov 22.

Crovella et al. Biological Pathways Associated With the Development of Pulmonary Toxicities in Mesothelioma Patients Treated With Radical Hemithoracic Radiation Therapy: A Preliminary Study. Frontiers in Oncology | www.frontiersin.org 1 December 2021 | Volume 11 | Article 784081.

Crovella et al.Variant Enrichment Analysis to Explore Pathways Disruption in a Necropsy Series of Asbestos-Exposed Shipyard Workers Int. J. Mol. Sci. 2022, 23(21), 13628; https://doi.org/10.3390/ijms232113628

Some acronims are reported in a wrong way:

Line 125: EMT should be explained  here and not in Line 146, Line 251: please indicate here (OS), Line 276: explain PFS here,

Conclusions should be drawn at the end of each paragraph:

Few words of summary /conclusion should be added at the endo of each paragraph.

Author Response

This review deals with some molecular aspects of the pathogenesis of the very aggressive tumor of the mesothelium, that is malignant mesothelioma (MPM). It  analyzes the role plaid by hypoxia, cancer stem cells and extracellular vesicles in the progression of this tumor. Furthermore it envisages how these factors could be useful targets to fight MPM.

We thank the Reviewer for careful reading of the manuscript and for fruitful suggestions and comments.  Below point-by-point answers (A) to each comment (C). The English language has been revised by Dr.  David Abbott, native English speaker, who has been enclosed in the author list.

Apart from the interest that this topic undoubtedly has, reading this review is difficult. The english language should be extensively revised. Some sentences are incomplete or too complex to be clearly understood and should be improved. Some important aspects of the MPM biology are not  discussed and finally the text suffers of many orthografic errors. Examples are reported as follows:

C1 Orthografic errors: Line 23: macrovesicles, Line 95: di?, Line 109:…in animal models,  Line 109: epithelial MPM?, Line 113: identified chromosomal deletions,  Line 193: NF-kb, Line 243: epithelioid,  activation, Line 245: strongly, Line 254: tumor, Line 278 NF2 should be called either merlin or Neurofibromin (table 1),  Line 281: ….is known to be  critical, Line 284: an, Line  286: TNF-a activity, Line 295: disease, Line: 296: related, Line 298: either…or, Line: 312: therapeutic, Line: 313: maintaining, Line 322: behavior, Line 343: MPM-EVs, Line 400: together, Line 401: intervention.

A1. We really thank the Reviewer for careful revision of the text. Orthographic and typo errors have been revised

C2. Sentences/Paragraphs which should be made more clear:English language requires an extensive revision by a English Mother Tongue person. However the following sentences should be made more understandable. Lines 84-86; Lines 98-100; Lines 157-160, Lines 170-178; Lines 183-195; Lines 251-259; Lines 267-270; Lines 309-311.

A2. We thank the Reviewer for this suggestion and revised the text accordingly. Dr Abbott, native English speaker revised the manuscript as well.

C3. References: missing or reported in a wrong way: Line 69: (16-20), Line 78: ..in the presence of genetic predisposition (missing), Line 119: missing, Line 220: (76-78), Line 228: (81-87), Line 251: (102-104), Line 270: Bronte??, Line 280: 119-121, Line 284: (126). The work of   Ito et al. (2021)  should be mentioned in the paragraph starting at Line 315, (Ito, F. et al. 2021. Ferroptosis-dependent extracellular vesicles from macrophage contribute to asbestos-induced mesothelial  carcinogenesis through loading ferritin. Redox Biol 47:102174. doi:10.1016/j.redox.2021.102174). Line 390: (monaco??).

A3. We really thank the Reviewer for careful revision of the text. Missing references have been added. In some cases it was possible to shorten the list since we used endnote to insert references.

C4.Important aspects of the MPM biology which are not discussed: The work of Crovella et al, Celsi et al , Borelli et al should be discussed.

Borelli et al. NLRP1 and NLRP3 polymorphisms in mesothelioma patients and asbestos exposed individuals a population-based autopsy study from North East Italy Infectious Agents and Cancer volume 10, Article number: 26 (2015) 

Crovella et al.  Iron signature in asbestos-induced malignant pleural mesothelioma: A population-based autopsy study. J Toxicol Environ Health A. 2016 Jan 28:1-13.

F Celsi et al. Pleural mesothelioma and lung cancer: the role of asbestos exposure and genetic variants in selected iron metabolism and inflammation genes. J Toxicol Environ Health A. 2019;82(20):1088-1102. doi: 10.1080/15287394.2019.1694612. Epub 2019 Nov 22.

Crovella et al. Biological Pathways Associated With the Development of Pulmonary Toxicities in Mesothelioma Patients Treated With Radical Hemithoracic Radiation Therapy: A Preliminary Study. Frontiers in Oncology | www.frontiersin.org 1 December 2021 | Volume 11 | Article 784081.

Crovella et al.Variant Enrichment Analysis to Explore Pathways Disruption in a Necropsy Series of Asbestos-Exposed Shipyard Workers Int. J. Mol. Sci. 2022, 23(21), 13628; https://doi.org/10.3390/ijms232113628

A4. We really thank the Reviewer for this suggestion; the text has been implemented as follows: “Great efforts have been directed to detect genetic susceptibility to asbestos cancerogenic potential focusing on genes involved in inflammatory infiltration, oxidative stress, chromosome instability and response to treatments []. In this perspective it has been reported that specific variants in three genes associated to iron metabolism, namely ferritin, transferrin, and hephaestin are significantly associated with protection against development of MPM []. Moreover, different pathway signatures have been detected in MPM samples from patients in response to tissue damages after lung-sparing surgery and chemotherapy and radical hemi-thoracic radiotherapy treatment with curative intent []. Due to their role in inflammasome, polymorphisms of NLRP1 and NLRP3 genes have been hypothesized to be involved in determining genetic susceptibility to MPM []. However preliminary results have not been validated in a cohort of MPM cases and controls with known asbestos exposure from Northern Italy and thus further validation are required []”.  

C5. Some acronims are reported in a wrong way: Line 125: EMT should be explained  here and not in Line 146, Line 251: please indicate here (OS), Line 276: explain PFS here,

A5.  We really thank the Reviewer for careful revision of the text. We revised the text has suggested.

C6. Conclusions should be drawn at the end of each paragraph: Few words of summary /conclusion should be added at the endo of each paragraph.

A6. We really thank the Reviewer for this suggestion. The text has been modified accordingly, as follows:

Section 2: In conclusion, it is well known that MPM has a uniquely poor somatic mutational landscape, and that the disease is mainly driven by microenvironmental selective pressure. Thus, available therapies are not MPM-specific, and patients’ outcome is still very poor and modestly influenced by current treatments. The absence of driving somatic lesions reflects clinical failure of small molecules, whereas immunotherapy has shown limited advantages in association with standard chemo. Growing interest is addressed to the germline MPM profile. A better understanding of MPM patients’ genetic variants and polymorphisms will contribute to decipher a still unexplored milieu and improve mechanistic knowledge on MPM biology and interindividual susceptibility to asbestos. Moreover, it could be of help in identifying novel actionable targets and in designing personalized and more efficient therapeutic strategies

Section 3 : In conclusion, as in other cancers, hypoxia deeply characterizes the MPM surrounding stroma and is strongly associated to tumor onset and progression. Moreover it is implicated in the failure of standard treatments. A deeper understanding of the molecular basis of the hypoxic mesothelioma microenvironment could be of help also in clinical setting.

Section 4: In conclusion, hypoxia represents a key feature involved in MPM onset and progression and it is strictly related to the maintenance of cancer cell hierarchical compartemnts. However the increasing body of knowledge regarding this peculiar bio-molecular context could be of help in designing tailored and more efficient therapeutic platforms.

Section 5: It is thus overall clear that EVs play a significant role in the development and progression of MPM []. Most importantly, EVs not only represent a key mechanistic system in MPM, but also identy a poweful actionable target. Further understanding of the mechanisms by which EVs contribute to cancer will pave the way for creating new diagnostic tools and therapeutic strategies.

Reviewer 2 Report

Dear Editor,
I have read with great interest the manuscript entitled The genes-stemness-secretome interplay in MPM: molecular dynamics and clinical hints. The Authors review the complex manuscript in biological mechanisms of MPM focusing on genetic, epigenetic alterations. Authors also explore resistance mechanisms. I believe that the review is well designed and written. Tables are very well presented. Minor points:
- I would suggest to discuss this recent article exploring the role of biological resistant mechanisms to anti pd-1 and anti VEGF agents  https://pubmed.ncbi.nlm.nih.gov/36669143/ 
- Please note some typos (reference 166 is written with a different character; a following reference is not numbered "Monaco").
Acceptable after these changes.

Author Response

I have read with great interest the manuscript entitled The genes-stemness-secretome interplay in MPM: molecular dynamics and clinical hints. The Authors review the complex manuscript in biological mechanisms of MPM focusing on genetic, epigenetic alterations. Authors also explore resistance mechanisms. I believe that the review is well designed and written. Tables are very well presented. Minor points:
- I would suggest to discuss this recent article exploring the role of biological resistant mechanisms to anti pd-1 and anti VEGF agents  https://pubmed.ncbi.nlm.nih.gov/36669143/ 
- Please note some typos (reference 166 is written with a different character; a following reference is not numbered "Monaco").

Reviewer 3 Report

The Review by Stella et al., entitled “The genes-stemness-secretome interplay in MPM: molecular dynamics and clinical hints”, discusses the novel therapeutic strategies focused on exploitation of MPM genetic asset and its interconnection with surrounding hypoxic microenvironment as well as transcript products and microvesicles.

Although it is well expanded and the topic is interesting, there are major text improvements to do facilitating understanding and fluidity of the work.

I would like to suggest to the authors pay more attention to manuscript preparation. A lot of spelling and typing errors compare to the main text, synonymous with haste and carelessness during the preparation of the manuscript. A careful proof-reading of the text is always recommended before submitting.

1.      There are some typing/spelling mistakes and authors are advised to carefully proof-read the text. For example,

-  in line number 23, the word “iacrovesicles” may be as “microvesicles”

-  in line number 107, “huma” as “human”;

- in line number 233, “detecteble” as “detectable”;

- in line number 239, “psudopodia” as “pseudopodia”;

- in line number 242, “aggressvenes” as “aggressiveness”;

- in line number 243, “eptheliod” as “epithelioid”, and “actiavtion” as “activation”;

- in line number 245, “strogly” as “strongly”;

- in line number 246, “immunosppression” as “immunosuppression”;

- in line number 270, “developped” as “developed”;

- in line number 295, “diasese” as “disease”;

- in line number 296, “realted” as “related”;

- in line number 305, “prognisis” as “prognosis”;

- in line number 312, “tehrapeutic” as “therapeutic”;

-in line number 396, “micronevironemntal” as “microenviromental”;

- in line number 398, “maintining” as “maintaining”.

The typos not mentioned here are also to be checked and corrected properly.

2.      The English need improvement since there are some grammatical and syntax errors in the manuscript. For example,

-  in line number 95, “The need to improve di accuracy” may be as “The need to improve accuracy”;

- in line number 281 “Notably merlin is known to critical for maintaining normal…” as “Notably, merlin is known to be critical for maintaining normal…”;

- lines number 281-283 please, rewrite the sentence or divide it into two separate sentences with relative references.

- in line number 284, delete “a”;

- in line number 285, write in italics only “Streptomyces plicatus”;

- in line number 313, “miRNAs has been reported to be involved in mainning the stemness” may be as “miRNAs have been reported to be involved in maintaining the stemness”;

- lines number 399-402 “The latter togher with exososmes and microvescicles, although behaving as key pathogenic clues, also define a different line of therapeutic intervations against MPM featuring a powerful translational potential for next future clinical development and use.” as “The latter together with exosomes and microvesicles, although behaving as key pathogenic clues, also define a different line of therapeutic intervention against MPM featuring a powerful translational potential for next future clinical development and use.”

The grammar mistakes which are not mentioned here are also to be checked and corrected properly.

3.      References errors to correct properly:

- in line number 106, please, use the correct formatting for inserting reference 27;

- in line number 119, please, enter the appropriate reference;

- in line number 190, please, enter the appropriate reference;

- in line number 270, please, enter the appropriate references;

- in line number 273, please, add a reference here;

- in line number 290, please, add a reference here;

- in lines number 309-311, please, add a reference here;

- in lines number 322-324, please, add a reference here;

- in line number 326, please enter the appropriate cited reference;

- in line number 335, please, add a reference here;

- in line number 349, please, move the reference outside the comma;

- in line number 377, please move this reference to the previous sentence (line number 375);

- in line number 387, please, use the correct formatting for inserting reference 166;

- in line number 390, please, enter the appropriate reference.

Author Response

The Review by Stella et al., entitled “The genes-stemness-secretome interplay in MPM: molecular dynamics and clinical hints”, discusses the novel therapeutic strategies focused on exploitation of MPM genetic asset and its interconnection with surrounding hypoxic microenvironment as well as transcript products and microvesicles. Although it is well expanded and the topic is interesting, there are major text improvements to do facilitating understanding and fluidity of the work. I would like to suggest to the authors pay more attention to manuscript preparation. A lot of spelling and typing errors compare to the main text, synonymous with haste and carelessness during the preparation of the manuscript. A careful proof-reading of the text is always recommended before submitting.

We thank the Reviewer for careful reading of the manuscript and for fruitful suggestions and comments.  Below point-by-point answers (A) to each comment (C). The English language has been revised by Dr.  David Abbott, native English speaker, who has been enclosed in the author list.

C1.      There are some typing/spelling mistakes and authors are advised to carefully proof-read the text. For example,

-  in line number 23, the word “iacrovesicles” may be as “microvesicles”

-  in line number 107, “huma” as “human”;

- in line number 233, “detecteble” as “detectable”;

- in line number 239, “psudopodia” as “pseudopodia”;

- in line number 242, “aggressvenes” as “aggressiveness”;

- in line number 243, “eptheliod” as “epithelioid”, and “actiavtion” as “activation”;

- in line number 245, “strogly” as “strongly”;

- in line number 246, “immunosppression” as “immunosuppression”;

- in line number 270, “developped” as “developed”;

- in line number 295, “diasese” as “disease”;

- in line number 296, “realted” as “related”;

- in line number 305, “prognisis” as “prognosis”;

- in line number 312, “tehrapeutic” as “therapeutic”;

-in line number 396, “micronevironemntal” as “microenviromental”;

- in line number 398, “maintining” as “maintaining”.

The typos not mentioned here are also to be checked and corrected properly.

A1. We thank the Reviewer for careful revision of our work. Typo errors have been revised.

C2.      The English need improvement since there are some grammatical and syntax errors in the manuscript. For example,

-  in line number 95, “The need to improve di accuracy” may be as “The need to improve accuracy”;

- in line number 281 “Notably merlin is known to critical for maintaining normal…” as “Notably, merlin is known to be critical for maintaining normal…”;

- lines number 281-283 please, rewrite the sentence or divide it into two separate sentences with relative references.

- in line number 284, delete “a”;

- in line number 285, write in italics only “Streptomyces plicatus”;

- in line number 313, “miRNAs has been reported to be involved in mainning the stemness” may be as “miRNAs have been reported to be involved in maintaining the stemness”;

- lines number 399-402 “The latter togher with exososmes and microvescicles, although behaving as key pathogenic clues, also define a different line of therapeutic intervations against MPM featuring a powerful translational potential for next future clinical development and use.” as “The latter together with exosomes and microvesicles, although behaving as key pathogenic clues, also define a different line of therapeutic intervention against MPM featuring a powerful translational potential for next future clinical development and use.”

The grammar mistakes which are not mentioned here are also to be checked and corrected properly.

A2. We thank the Reviewer for careful revision of the text. Typo errors have been revised as indicated. English language has been revised by Dr Abbott, native English speaker.

C3.      References errors to correct properly:

- in line number 106, please, use the correct formatting for inserting reference 27;

- in line number 119, please, enter the appropriate reference;

- in line number 190, please, enter the appropriate reference;

- in line number 270, please, enter the appropriate references;

- in line number 273, please, add a reference here;

- in line number 290, please, add a reference here;

- in lines number 309-311, please, add a reference here;

- in lines number 322-324, please, add a reference here;

- in line number 326, please enter the appropriate cited reference;

- in line number 335, please, add a reference here;

- in line number 349, please, move the reference outside the comma;

- in line number 377, please move this reference to the previous sentence (line number 375);

- in line number 387, please, use the correct formatting for inserting reference 166;

- in line number 390, please, enter the appropriate reference.

A3. We thank the Reviewer for careful revision of the text. References have been revised accordingly.

Round 2

Reviewer 1 Report

None

Reviewer 3 Report

The manuscript has been properly finalized, following the required revisions.